# Ground-Compose-Reinforce: Grounding Language in Agentic Behaviours using Limited Data

**Andrew C. Li, Toryn Q. Klassen[†], Andrew Wang, Parand A. Alamdari[†], Sheila A. McIlraith[†]**
Department of Computer Science, University of Toronto
Vector Institute for Artificial Intelligence
[†] Schwartz Reisman Institute for Technology and Society
Toronto, Canada
{andrewli,toryn,andrewwang,parand,sheila}@cs.toronto.edu

## Abstract

Grounding language in perception and action is a key challenge when building situated agents that can interact with humans, or other agents, via language. In the past, addressing this challenge has required manually designing the language grounding or curating massive datasets that associate language with the environment. We propose Ground-Compose-Reinforce, an end-to-end, neurosymbolic framework for training RL agents directly from high-level task specifications—without manually designed reward functions or other domain-specific oracles, and without massive datasets. These task specifications take the form of *Reward Machines*, automata-based representations that capture high-level task structure and are in some cases autoformalizable from natural language. Critically, we show that Reward Machines can be grounded using limited data by exploiting compositionality. Experiments in a custom Meta-World domain with only 350 labelled pretraining trajectories show that our framework faithfully elicits complex behaviours from high-level specifications—including behaviours that never appear in pretraining—while non-compositional approaches fail.

## 1 Introduction

Grounding language—connecting language with perception and action within an environment—is a fundamental challenge when building robots and other agents that are interfaced through language. One popular approach to addressing this challenge is to employ a *manually-designed* domain-specific interpretation of language, such as a language-conditional reward function or success detector (e.g. [1–4]). For instance, in the BabyAI benchmark [1], successful execution of instructions like "go to the red ball" can be evaluated programmatically in the environment simulator, providing a reward signal for learning language-conditioned behaviours. Such instances of grounded language generalize to arbitrary scenarios and controlled subsets of language by design, but are hard to hand-engineer in complex, non-simulated settings based on raw perceptual inputs like pixels.

The recent advent of large language models (LLMs) has inspired an alternative pathway to grounding language: training on diverse datasets that pair language descriptions with environment trajectories (e.g., $\pi_0$ [5], RT-2 [6], LIV [7], VPT [8]). While this obviates the need for manually designed language groundings, it typically requires enormous datasets in order to capture the broad scope of language usage within an environment [1, 9, 10]. For agentic applications that are data-intensive (e.g. robotics) or where access to trajectory data is limited, such data-driven language models are prone to failure on complex or out-of-distribution tasks [11–14].

We propose **Ground-Compose-Reinforce**, an end-to-end framework for training reinforcement learning (RL) agents directly from high-level task specifications, *without* relying on manually

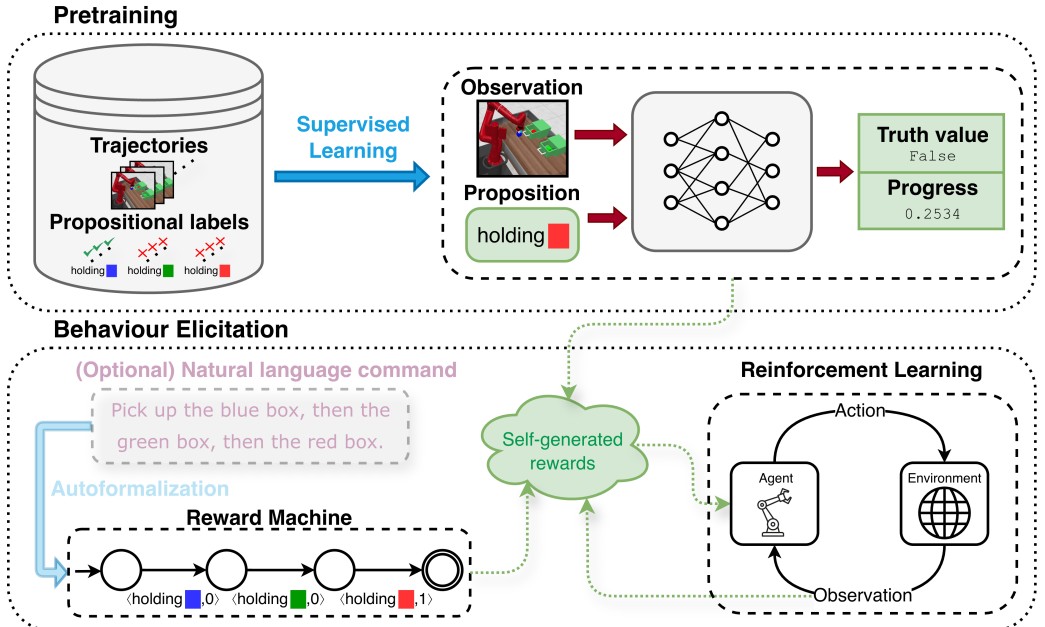

Figure 1: Ground-Compose-Reinforce, a lightweight framework for training RL agents directly from Reward Machine specifications, without oracles like reward functions or feature detectors. In pretraining, we learn to map propositional symbols to context-specific truth values ("is the robot holding the red block?") and progress signals ("how close is the robot to holding the red block?"). To elicit behaviours, we prompt the agent via a Reward Machine composed of these symbols (or via natural language, if an autoformalizer is available). The agent then synthesizes its *own* dense reward function and interacts with the environment to learn the desired behaviour via RL.

designed rewards or feature detectors. We represent tasks via *Reward Machines* (RMs) [15, 16], an automata-based task specification language that exposes temporally extended task structure in terms of a set of atomic propositional symbols. Specifically, our framework: (1) **grounds** these atomic symbols in the environment by pretraining on a limited dataset of labelled trajectories; (2) **composes** these symbols via RMs to express complex tasks; and (3) for any such task, trains an agent to solve that task via **RL** on self-generated dense rewards based on its learned interpretation of the RM. Overall, Ground-Compose-Reinforce enables the elicitation of a wide range of temporally and logically structured behaviours expressed as RMs (specified directly or in some cases autoformalized from natural language), requires minimal pretraining data, and does not rely on external, domain-specific oracles for training or execution. In this paper, we present the following novel contributions:

1. A conceptual, end-to-end framework for compositionally grounding language in behaviours based on Reward Machines and reinforcement learning (Figure 1). Critically, our framework requires minimal labelled trajectory data for pretraining, and does not require external oracles like reward functions, success detectors, or feature detectors.

2. A compositional reward shaping strategy for Reward Machine tasks that is critical for strong performance in Meta-World [17]. Our strategy addresses *propositional sparsity*, where propositions of interest (e.g. "pick up the block") are rarely satisfied through random exploration.

3. Experiments across diverse Reward Machine tasks, including temporally extended gridworld navigation and robotic manipulation in Meta-World. Our approach elicits diverse behaviours in Meta-World (including out-of-distribution behaviours that never appear in the pretraining dataset) from only 350 labelled pretraining trajectories while non-compositional approaches fail.

## 2   Related Work

**Grounded Language Learning.** Several past works have explored grounded language learning. Hermann et al. [2] show that an RL agent in a 3-D environment can consistently navigate to target

objects described via language by being rewarded for successful trajectories. Hill et al. [3] show that an agent can learn new word-object bindings and apply that knowledge to solve tasks within the same episode. Liu et al. [18] show that a meta-RL agent indirectly learns to interpret language-based advice embedded within an environment to improve its task performance. Chaplot et al. [4] propose a neural architecture for grounded language learning in a 3-D Doom environment, considering both RL with success-based rewards and imitation learning from an oracle policy. Chevalier-Boisvert et al. [1] propose the BabyAI platform for learning a synthetic language in a 2-D gridworld, finding that existing RL and imitation learning approaches are sample inefficient and generalize poorly. Unlike our approach, these methods require significant manual design of reward functions or oracle policies.

Recent works learn grounded language from data, without domain-specific manual design. Black et al. [5], Zitkovich et al. [6] and Kim et al. [19] present vision-language-action models for robotics while Baker et al. [8] and Lifshitz et al. [13] present models for playing MineCraft. Bahdanau et al. [20] and Ma et al. [7] learn language-conditioned reward functions from data. Works have also directly leveraged vision-language models for rewards in vision-based environments [21–23], but typically do not leverage language compositionality and are prone to failure on complex or out-of-distribution tasks. Shi et al. [12], Yuan et al. [24], Huang et al. [25] and Ahn et al. [26] decompose complex tasks at execution time via language models. While this shares motivation with our work, we represent tasks via RMs and exploit compositional task structure for both training and execution.

**Formal Languages for RL.** Formal languages like Linear Temporal Logic (LTL) [27] and other associated formal structures[1] such as RMs [16] have a rich history of application in the control [28, 29], verification [30–32], monitoring, and synthesis of dynamical systems. Recently, they have risen in popularity in deep RL for white-box specification of rich temporally extended reward functions [15, 33–35] and in a number of cases can be automatically generated from natural language commands (e.g., [36–39]). Several works show that formal languages enable compositional generalization to unseen instructions. Vaezipoor et al. [40], Kuo et al. [41] and Yalcinkaya et al. [42] train instruction-conditioned policies that zero-shot generalize to unseen instructions by training on procedurally generated LTL formulas. Qiu et al. [43], Liu et al. [44], León et al. [45] and Jackermeier and Abate [46] train transferable skills that can be invoked by a planner. Nangue Tasse et al. [47] consider how value functions and policies can be composed zero-shot for arbitrary RM tasks, but assume that all the tasks can be captured via a finite, predetermined set of goal states. Our compositional reward shaping approach also builds on prior methods. Camacho et al. [34], Furelos-Blanco et al. [48] and Parać et al. [49] provide potential-based rewards for RMs based on the current RM state, but such methods provide no signal when target propositions rarely occur. Several works propose continuous progress signals for each proposition [33, 50–54], but these approaches are limited to tasks with a binary success criterion. While progress signals are typically manually specified, we show how they can be learned directly from data.

Nearly all formal-language-based deep RL methods assume access to an external evaluator of symbolic features (a.k.a. a labelling function). To our knowledge, only a few works specifically avoid this assumption, but they instead depend on an external reward signal. Li et al. [55, 56] consider the implications of noisily grounding symbols when using RMs. Hyde and Santos [57] and Christoffersen et al. [58] infer RMs from the reward signal as an inductive bias for RL. Umili et al. [59, 60], Kuo et al. [41], Andreas et al. [61] and Oh et al. [62] show that *ungrounded* formal specifications can improve RL by providing information about the task structure. In contrast to these works, our approach does not rely on an external symbol evaluator or reward function.

## 3 Preliminaries

### 3.1 Reinforcement Learning

A reinforcement learning (RL) problem considers an environment modelled as a *Markov Decision Process* (MDP) $\langle \mathcal{S}, \mathcal{A}, \mathcal{T}, \mathcal{P}, R, \mu, \gamma \rangle$, where $\mathcal{S}$ is a set of states, $\mathcal{A}$ a set of actions, $\mathcal{T} \subset \mathcal{S}$ a set of terminal states, $\mathcal{P} : \mathcal{S} \times \mathcal{A} \to \Delta\mathcal{S}$ a transition probability distribution, $R : \mathcal{S} \times \mathcal{A} \times \mathcal{S} \to \mathbb{R}$ a reward function, $\mu \in \Delta\mathcal{S}$ an initial probability distribution, and $\gamma \in [0, 1]$ a discount factor. An episode begins with $s_0 \sim \mu$, and at each time $t \geq 0$ the agent chooses an action $a_t$, then observes the next state $s_{t+1} \sim \mathcal{P}(s_t, a_t)$ and reward $r_{t+1} = R(s_t, a_t, s_{t+1})$, repeating until a terminal state

---

[1]Henceforth "formal languages," for ease of exposition.

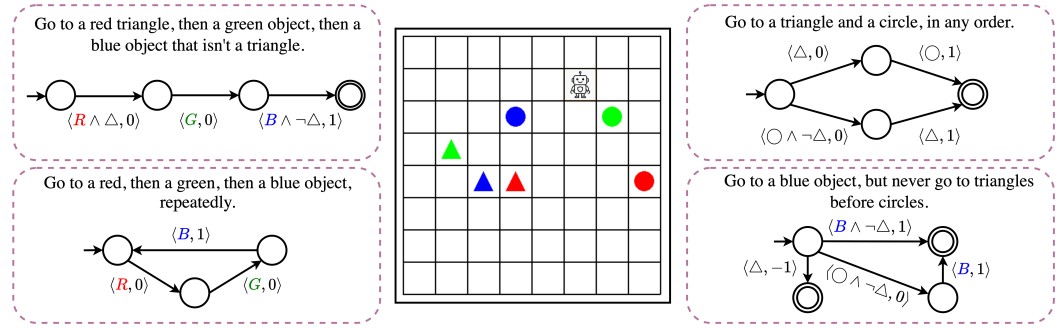

Figure 2: Four temporally extended tasks in a gridworld expressed as Reward Machines over the propositions $\mathcal{AP} = \{R, G, B, \triangle, \bigcirc\}$. An edge labelled $\langle \varphi, r \rangle$ indicates the logical condition $\varphi$ for when the corresponding transition should be followed, and the reward $r$ that is yielded as a result. Doubled circles indicate terminal states, and we omit non-rewarding self-loop edges to aid readability.

$s_T \in \mathcal{T}$ is reached. We refer to full episodes as *trajectories*, denoted by $\tau = \langle s_0, a_0, s_1, a_1, \ldots \rangle$. A *history* $h_t = \langle s_0, a_0, s_1, a_1, \ldots s_t \rangle$ refers to the states and actions up to time $t$. The agent's goal is to interact with the environment to learn a policy $\pi(a_t|s_t)$ that maximizes the *expected discounted return* $\mathbb{E}_{\tau \sim \pi}[\sum_{t=1}^{T} \gamma^t r_t]$ (where the episode length $T$ can be $\infty$).

## 3.2 Task Specification via Reward Machines

Formal languages with sequential or temporal structure—including RMs [16], *regular expressions* [63], and various *temporal logics* [54, 40]—offer an intuitive and expressive interface for specifying tasks in RL while supporting representations that capture compositional task structure. Starting with a predefined set of *atomic propositions* $\mathcal{AP}$ that represent abstract, binary features of environment states, such languages can be used to express temporally extended properties or reward functions. In this work, we focus on tasks specified as RMs, which subsume several formal languages of interest [34] such as LTL over finite traces [64, 65] and regular expressions.

An RM is an automaton that captures the structure of a reward function over the abstract vocabulary $\mathcal{AP}$ (Figure 2). It takes as input a sequence of *truth assignments* $\langle \omega_1, \omega_2, \ldots \rangle$ over $\mathcal{AP}$, where each $\omega_t \in 2^{\mathcal{AP}}$ denotes the subset of $\mathcal{AP}$ that holds true at time $t$, and outputs a corresponding sequence of rewards $\langle r_1, r_2, \ldots \rangle$. An RM has a finite set of internal states $\mathcal{U}$ and begins in a fixed state $u_0$ at time $t = 0$. At each step $t \geq 1$, the RM updates its state to $u_t \in \mathcal{U}$ and emits a reward $r_t$ based on the current input $\omega_t$ and the previous state $u_{t-1}$. This continues until the RM enters a terminal state from a designated subset $\mathcal{F} \subset \mathcal{U}$.

**Running Example.** *The gridworld in Figure 2 will serve as a running example. Consider the top-left RM, which describes a task composed of three subgoals to be completed in a fixed order. When given an input sequence $\langle \omega_1, \omega_2, \ldots \rangle$ that identifies the values of* all *propositions at each time $t$ (whether the agent is at a red object, a green object, a circle, and so on), the RM state $u_t \in \mathcal{U}$ tracks which subgoals have been completed and transitions to the next RM state as soon as the agent achieves the current subgoal. When a red triangle, a green object, and a blue object that isn't a triangle are reached in that order, the RM terminates with a reward of 1.*

**Definition 1.** *A **Reward Machine** $\mathcal{R}$ is defined as a tuple $\langle \mathcal{U}, u_0, \mathcal{F}, \mathcal{AP}, \delta_u, \delta_r \rangle$, where $\mathcal{U}$ is a finite set of states, with initial state $u_0 \in \mathcal{U}$ and terminal states $\mathcal{F} \subset \mathcal{U}$; $\mathcal{AP}$ is a set of propositions; $\delta_u : \mathcal{U} \times 2^{\mathcal{AP}} \to \mathcal{U}$ is a transition function that updates the RM state based on the current truth assignment; and $\delta_r : \mathcal{U} \times 2^{\mathcal{AP}} \to \mathbb{R}$ is a reward function that emits a reward at each step.*

As shown in Figure 2, the transition and reward functions of an RM can be intuitively and compactly specified via a set of labelled edges of the form $\langle u, u', \varphi, r \rangle$, indicating that if a truth assignment $\omega$ satisfies the formula $\varphi$ (denoted by $\omega \models \varphi$), then $\delta_u(u, \omega) = u'$ and $\delta_r(u, \omega) = r$.

## 3.3 Grounded Interpretations

RMs express tasks in terms of abstract symbols (e.g., $R$, $\triangle$), but these symbols must be *grounded* in the environment to be meaningful. This is achieved via a *labelling function* $\mathcal{L} : \mathcal{S} \to 2^{\mathcal{AP}}$ that

maps each MDP state $s$ to the set $\omega \subseteq \mathcal{AP}$ of propositions that hold true in $s$. For an RM $\mathcal{R}$ over propositions $\mathcal{AP}$, any such $\mathcal{L}$ also grounds $\mathcal{R}$ in the environment by inducing a reward sequence for any MDP trajectory $\tau$: first, states $s_t$ in $\tau$ are converted into truth assignments $\omega_t = \mathcal{L}(s_t)$, then the RM is simulated over the sequence $\langle \omega_1, \omega_2, \ldots \rangle$ to generate a sequence of rewards until termination.

An *RM-MDP* augments an environment (represented by a reward-free MDP) with a concrete reward function captured by an RM and labelling function. The resulting reward function is generally *non-Markovian* with respect to MDP states $\mathcal{S}$, since the reward at time $t$ depends on the internal RM state $u_t$. While one might consider expressing optimal behaviours via a history-based policy $\pi(a_t|h_t)$, this is unnecessary if the agent has oracle access to $\mathcal{L}$ since RM-MDPs are Markovian over the extended state space $\mathcal{S} \times \mathcal{U}$ [16]. The agent can use $\mathcal{L}$ to compute $\omega_t = \mathcal{L}(s_t)$ and recursively simulate the RM to track $u_t$. Thus, it is typical to express policies in the form $\pi(a_t|s_t, u_t)$, where the RM state $u_t$ compactly encodes the history $h_t$ and is sufficient for optimal decision making.

**Definition 2.** *An **RM-MDP** is a triple $\langle \mathcal{M}, \mathcal{R}, \mathcal{L} \rangle$, where $\mathcal{M} = \langle \mathcal{S}, \mathcal{A}, \mathcal{P}, \mu, \gamma \rangle$ is an MDP without rewards or terminal states, $\mathcal{R} = \langle \mathcal{U}, u_0, \mathcal{F}, \mathcal{AP}, \delta_u, \delta_r \rangle$ is an RM, and $\mathcal{L} : \mathcal{S} \to 2^{\mathcal{AP}}$ is a labelling function. The RM-MDP is equivalent to an MDP with state space $\mathcal{S} \times \mathcal{U}$ and reward function induced by $\mathcal{R}$ and $\mathcal{L}$.*

**Running Example.** *The RM at the top left of Figure 2 captures the high-level structure of a multi-stage task. To map environment trajectories into concrete rewards for this RM, we need a* labelling function $\mathcal{L} : \mathcal{S} \to 2^{\mathcal{AP}}$ *that connects abstract propositions like* R *and* △ *to environment states.*

# 4 Problem Setting

Our goal in this work is to faithfully elicit behaviours from an agent given only a high-level task specification in the form of an RM, $\mathcal{R}$, such as the ones depicted in Figure 2. $\mathcal{R}$ can be specified directly, translated from other formal languages like LTL [34], generated from a symbolic planner [66, 67], or sometimes autoformalized from natural language. Formally, we consider an environment $\mathcal{M} = \langle \mathcal{S}, \mathcal{A}, \mathcal{P}, \mu, \gamma \rangle$ (an MDP without rewards or terminal states) and a finite set of propositional symbols $\mathcal{AP}$. For any given RM task $\mathcal{R} = \langle \mathcal{U}, u_0, \mathcal{F}, \mathcal{AP}, \delta_u, \delta_r \rangle$, we wish to obtain a policy $\pi_{\mathcal{R}}(a_t|h_t)$ that performs well in the RM-MDP $\langle \mathcal{M}, \mathcal{R}, \mathcal{L}^* \rangle$, where $\mathcal{L}^* : \mathcal{S} \to 2^{\mathcal{AP}}$ reflects a ground-truth interpretation of the propositions $\mathcal{AP}$ in the environment.[2]

**Assumptions.** We aim to obtain $\pi_{\mathcal{R}}$ *without* online access to $\mathcal{L}^*$ or to an external reward function that evaluates ground-truth performance with respect to $\mathcal{R}$.[3] In order to connect symbols $\mathcal{AP}$ with environment percepts, we instead assume access to a fixed pretraining dataset $\mathcal{D} = \{\langle \tau^i, \omega^i \rangle\}_{i=1}^N$ of trajectories $\tau^i = \langle s_0^i, a_0^i, s_1^i, \ldots \rangle$ with corresponding labels $\omega^i = \langle \mathcal{L}^*(s_0^i), \mathcal{L}^*(s_1^i), \ldots \rangle$. In practice, such labels can be obtained via crowdsourced annotations [69] or self-supervised learning [70].

For a task $\mathcal{R}$, the agent is allowed an arbitrary number of interaction episodes with the environment before committing to a final policy $\pi_{\mathcal{R}}$. However, during this interaction phase, the agent must learn in a self-supervised manner as the environment does not provide a separate reward signal.

# 5 Ground-Compose-Reinforce

We propose an end-to-end framework for this setting called *Ground-Compose-Reinforce* (Figure 1). In the pretraining phase, the agent first grounds propositional symbols $\mathcal{AP}$ in environment states via supervised learning on $\mathcal{D}$. In the behaviour elicitation phase, the agent is given a task as an RM $\mathcal{R}$ composed over symbols $\mathcal{AP}$. The agent then learns a policy $\pi_{\mathcal{R}}$ by interacting with the environment and synthesizing its own learning signal for RL based on $\mathcal{R}$ and its learned interpretation of $\mathcal{AP}$. We hypothesize that this bottom-up approach to grounding language in behaviours—first learning the meanings of individual symbols and then composing them to interpret complex tasks—is an effective strategy. The remainder of this section describes the core implementation of Ground-Compose-Reinforce and in Section 6, we raise and address an issue called *propositional sparsity* where the agent fails to learn in extremely long-horizon tasks.

---

[2]For the problem statement, we consider the more general form of history-based policies, rather than policies conditioned on the ground-truth RM state, which depend on $\mathcal{L}^*$.

[3]Such oracles can be notoriously hard to design in practice [68] and often require internal simulator access.

**Core Algorithm.** We ground the propositional symbols $\mathcal{AP}$ by learning a labelling function $\hat{\mathcal{L}}(s) \approx \mathcal{L}^*(s)$ via any binary classification method on $\mathcal{D}$. Given an RM task $\mathcal{R}$ over $\mathcal{AP}$, we solve a *surrogate* RM-MDP $\langle \mathcal{M}, \mathcal{R}, \hat{\mathcal{L}} \rangle$ via RL. This surrogate task approximates the true RM-MDP $\langle \mathcal{M}, \mathcal{R}, \mathcal{L}^* \rangle$ for which the agent lacks supervision. Since the agent can query $\hat{\mathcal{L}}$ freely, it can simulate rewards $\langle r_1, r_2, \ldots \rangle$ and RM states $\langle u_1, u_2, \ldots \rangle$ for any trajectory, as described in Section 3.3. Finally, we use these self-generated signals to train a policy $\pi_{\mathcal{R}}(a_t | s_t, u_t)$ as outlined in Algorithm 1.

---

**Algorithm 1** Ground-Compose-Reinforce for RMs

---

**Input:** MDP $\mathcal{M}$ without rewards, Propositional symbols $\mathcal{AP}$, Dataset $\mathcal{D}$ of labelled trajectories, RM task $\mathcal{R}$ over $\mathcal{AP}$

    *// Pretraining phase*
1: Train labelling function $\hat{\mathcal{L}}(s)$ on $\mathcal{D}$ using any binary classification method
    *// Behaviour elicitation phase*
2: Initialize policy $\pi_{\mathcal{R}}(a \mid s, u)$ arbitrarily
3: **for** each episode **do**
4:     Observe initial state $s$ in $\mathcal{M}$; set $u$ to the initial state of $\mathcal{R}$
5:     **while** $u$ is non-terminal **do**
6:         Sample action $a \sim \pi_{\mathcal{R}}(\cdot \mid s, u)$
7:         Execute $a$ in $\mathcal{M}$ and observe next state $s'$
8:         Compute truth assignment $\hat{\omega} \leftarrow \hat{\mathcal{L}}(s')$
9:         Update RM: $u' \leftarrow \delta_u(u, \hat{\omega})$, $r \leftarrow \delta_r(u, \hat{\omega})$
10:        Update $\pi_{\mathcal{R}}$ with RL for transition $\langle s, u, a, r, s', u' \rangle$
11:        Set $s \leftarrow s'$, $u \leftarrow u'$

---

**Running Example.** *Suppose our gridworld agent is expected to solve arbitrary RM tasks over the vocabulary $\{\mathrm{R}, \mathrm{G}, \mathrm{B}, \triangle, \bigcirc\}$. With Ground-Compose-Reinforce, the agent first connects these symbols to environment states via $\mathcal{D}$ (i.e. it learns to identify which states have red shapes, which states have triangles, etc). A human can then specify a new task as an RM composed over these symbols (e.g. "go to a triangle and a circle, in any order") without needing to program a task-specific reward function. The agent systematically evaluates its own performance on this task by composing its learned interpretations of $\{\mathrm{R}, \mathrm{G}, \mathrm{B}, \triangle, \bigcirc\}$, providing a learning signal for RL.*

## 6 Compositional Reward Shaping

An issue that can arise with the core implementation is *sparse rewards*. While prior RM works have proposed additional learning signals for transitioning in the RM [34, 48, 49], these methods fail when RM transitions *themselves* pose an exploration challenge. This can be caused by *propositional sparsity*: when $\mathcal{L}^*(s)$ is constant across most states under random exploration, the agent struggles to learn to meaningfully affect propositional values (e.g. consider a robot manipulation task that involves satisfying the proposition "the box is picked up").

To address this, we extend Ground-Compose-Reinforce with *potential-based reward shaping* [71]. Specifically, for a given RM task $\mathcal{R}$, we estimate the *optimal value function* (OVF) for the surrogate RM-MDP $\langle \mathcal{M}, \mathcal{R}, \hat{\mathcal{L}} \rangle$ by decomposing $\mathcal{R}$ into simpler subtasks. The most basic subtasks correspond to satisfying individual propositions or their negations, and we estimate the OVFs for these $2|\mathcal{AP}|$ tasks during pretraining via offline RL on $\mathcal{D}$. These $2|\mathcal{AP}|$ OVFs

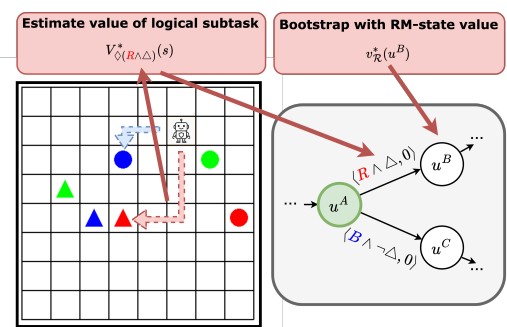

Figure 3: An illustration of how we estimate optimal values in an RM-MDP. Suppose the agent is currently in RM state $u^A$ (green and bolded). To evaluate the expected return for the transition $u^A \to u^B$, we estimate how close the agent is to satisfying the formula on the transition (reaching the red triangle), and bootstrap with a coarse value estimate for RM state $u^B$. The overall value of $u^A$ is approximated by the maximum expected return across all outgoing transitions from $u^A$.

then serve as building blocks that can be composed to approximate the OVF for *any* of the *infinitely many* RM tasks over $\mathcal{AP}$.

### 6.1 Deconstructing RMs into Logical Subtasks

To estimate the OVF of the surrogate RM-MDP $\langle \mathcal{M}, \mathcal{R}, \hat{\mathcal{L}} \rangle$, denoted $V_{\mathcal{R}}^*(s, u)$, we combine two key ideas: (1) treating each RM transition in $\mathcal{R}$ as an independent subtask, and (2) bootstrapping from a state-independent value function $v_{\mathcal{R}}^*(u)$. Consider the example in Figure 3, where the agent is in

MDP state $s$ and RM state $u^A$. Each RM transition is associated with a logical condition $\varphi$ (e.g., $R \wedge \triangle$ or $B \wedge \neg\triangle$), and we treat the satisfaction of $\varphi$ as its own subtask. Definition 3 formalizes this: we introduce the class $\Diamond\mathbf{PL}(\mathcal{AP})$ of reachability tasks over propositional formulas, where each task $\Diamond\varphi$ entails reaching a state $s_T$ such that $\hat{\mathcal{L}}(s_T) \models \varphi$, terminating with reward 1 upon satisfaction. $\Diamond\varphi$ is itself an RM-MDP with a single non-terminal RM state (and a terminal RM state), and is therefore Markovian over $\mathcal{S}$. We denote its OVF as $V^*_{\Diamond\varphi}(s)$.

**Definition 3.** *For any propositional logic formula $\varphi$ over $\mathcal{AP}$, define $\Diamond\varphi$ as the task of reaching a state $s_T \in \mathcal{S}$ such that $\hat{\mathcal{L}}(s_T) \models \varphi$. The episode terminates and yields a reward of 1 if such a state is reached, and continues indefinitely with 0 reward otherwise. Let $\Diamond\mathbf{PL}(\mathcal{AP}) = \{\Diamond\varphi \mid \varphi$ is a propositional formula over $\mathcal{AP}\}$ denote the set of such reachability tasks.*

The second component of our method is to bootstrap using a state-independent value function $v^*_{\mathcal{R}}(u)$ that approximates the expected return from any RM state $u$ while ignoring the MDP state. We estimate $v^*_{\mathcal{R}}(u)$ using a variant of Value Iteration over the RM graph, following Camacho et al. [34]. Finally, to estimate $V^*_{\mathcal{R}}(s, u)$, we evaluate each outgoing transition $\langle u, u', \varphi, r \rangle$ from $u$ by combining the subtask value $V^*_{\Diamond\varphi}(s)$ with the bootstrapped value of the next RM state $v^*_{\mathcal{R}}(u')$. The final estimate is the maximum expected return over all such transitions:

$$V^*_{\mathcal{R}}(s, u) \approx \max_{\langle u, u', \varphi, r \rangle} \left[ V^*_{\Diamond\varphi}(s) \cdot \left( r + \gamma v^*_{\mathcal{R}}(u') \right) \right] \tag{1}$$

This approximation assumes no RM self-transitions with non-zero rewards. Appendix A further justifies and explains our approximation while extending it to arbitrary RMs.

## 6.2 Further Deconstructing Logical Subtasks

Approximation 1 allows us to estimate $V^*_{\mathcal{R}}(s, u)$ for any RM task $\mathcal{R}$ over $\mathcal{AP}$, provided we can estimate $V^*_{\Diamond\varphi}(s)$ for any $\varphi$ in $\Diamond\mathbf{PL}(\mathcal{AP})$. However, the number of propositional formulas over $\mathcal{AP}$ (up to logical equivalence) is $2^{2^{|\mathcal{AP}|}}$ and modelling a separate OVF for each such task is intractable. To address this, we further decompose logical formulas based on their structure. Any formula $\varphi$ can be rewritten in disjunctive normal form, i.e., as a disjunction of conjunctions of literals (where a literal is either a proposition $x$ or its negation $\neg x$). We then approximate the OVF of $\Diamond\varphi$ using the semantics of fuzzy logic [72], where $\max$ represents disjunction and $\min$ represents conjunction.[4]

Let $\varphi = \xi_1 \vee \ldots \vee \xi_k$, where each $\xi_i$ is a conjunction of literals. We approximate:

$$V^*_{\Diamond\varphi}(s) \approx \max_{i=1,\ldots,k} V^*_{\Diamond\xi_i}(s) \tag{2}$$

For each conjunctive clause $\xi = l_1 \wedge \ldots \wedge l_k$, where each $l_i$ is a literal, we approximate:

$$V^*_{\Diamond\xi}(s) \approx \min_{i=1,\ldots,k} V^*_{\Diamond l_i}(s) \tag{3}$$

By composing Approximations 1–3, we can estimate the OVF for any RM task based on only $2|\mathcal{AP}|$ OVFs—namely, those for $\Diamond x$ and $\Diamond\neg x$, for each $x \in \mathcal{AP}$. We refer to these as the *primitive value functions* (PVFs).

## 6.3 Final Remarks

In this section, we showed that the optimal value function (OVF) of *any* RM task $\mathcal{R}$ can be approximated using just $2|\mathcal{AP}|$ *primitive value functions* (PVFs). From these $2|\mathcal{AP}|$ PVFs, we can estimate OVFs for *doubly exponentially many* logical tasks ($2^{2^n}$) and *infinitely many* RM tasks. Each PVF quantifies progress toward satisfying a single proposition or its negation, and can be learned directly from $\mathcal{D}$ using any offline RL algorithm. One might view this approach as trading off expressivity for modularity: directly modelling the OVFs of all RM tasks is infeasible, so we instead model a small, reusable set of $2|\mathcal{AP}|$ PVFs at the cost of introducing some approximation error.

Leveraging that we can estimate the OVF for any RM task $\mathcal{R}$, we extend Algorithm 1 with potential-based reward shaping to address propositional sparsity. Further details on learning PVFs, sources of approximation error, and the potential-based reward shaping scheme are provided in Appendix A.

---

[4]Fuzzy operators have previously been applied for satisfaction of a temporal formula over quantitative signals [33, 53, 54]. While such tasks are binary in nature, we consider RMs, which can express other reward structures.

# 7 Experiments

We conducted experiments to evaluate the following research questions:

**RQ1 Grounding RMs in Behaviours**: With Ground-Compose-Reinforce, can we faithfully elicit behaviours given high-level task specifications (RMs)?

**RQ2 Compositional Generalization**: Can we elicit meaningful, out-of-distribution (OOD) behaviours beyond those observed in $\mathcal{D}$?

**RQ3 Propositional Sparsity**: Can the agent operate in extremely long-horizon environments where propositional values are hard to alter with random exploration?

*Code/videos available at:* `https://github.com/andrewli77/ground-compose-reinforce`.

## 7.1 Experimental Setup

We considered two domains: an image-based gridworld with randomized object locations called *GeoGrid* (introduced in the running example), and a Meta-World-based robotics environment called *DrawerWorld* (Figure 4). Full details on the setup can be found in Appendix B.1.

**Pretraining Datasets.** We collected $\mathcal{D}$ under minimal assumptions about downstream tasks. In GeoGrid, $\mathcal{D}$ contains 5000 trajectories generated by a random policy. In DrawerWorld, we manually operated the robot to collect 350 trajectories involving generic behaviours (e.g., opening drawers, lifting boxes). To evaluate OOD generalization, we constrained $\mathcal{D}$ to only contain trajectories interacting with at most one box in DrawerWorld. Finally, trajectories were labelled using a handcrafted labelling function.

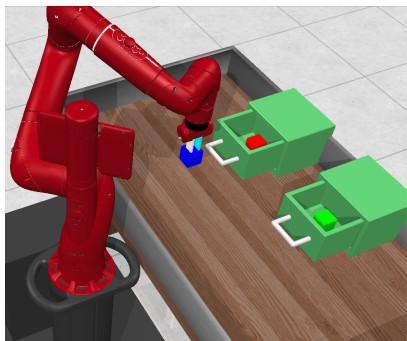

Figure 4: *DrawerWorld* is a custom Meta-World environment where the agent can interact with two drawers and three boxes. Propositions capture whether: each drawer is open; each box is lifted by the agent; a given box is in a given drawer.

**Tasks.** We designed a diverse set of RM tasks (Table 1) that target behaviours rarely or never seen in $\mathcal{D}$. The GeoGrid tasks evaluate whether the agent can produce fine-tuned behaviours beyond the random-action trajectories observed in $\mathcal{D}$. The DrawerWorld tasks evaluate whether the agent can solve complex manipulation tasks that require composing behaviours observed in $\mathcal{D}$ (e.g., *Pickup-Each-Box* requires handling all three boxes, while trajectories in $\mathcal{D}$ interact with at most one box).

## 7.2 Method and Baselines

We benchmarked Ground-Compose-Reinforce (GCR) against several non-compositional baselines. Methods based on online RL (GCR, Bespoke Reward Model) use PPO [73] to train a policy from scratch. During execution, GCR captures memory via the RM state while all non-RM-based baselines encode the observation history using an additional GRU [74]. See Appendix B.2 for full implementation details and Appendix B.3 for full training details.

**Ground-Compose-Reinforce (ours).** We implemented GCR with potential-based reward shaping as described in Sections 5 and 6. Both the predicted labelling function $\hat{\mathcal{L}}$ and PVFs are neural networks trained on $\mathcal{D}$ via supervised learning and offline RL, respectively.

**LTL-conditioned Behaviour Cloning (LTL-BC)** is a neural network policy $\pi(a_t|h_t, \varphi)$ that directly maps LTL specifications $\varphi$ to behaviours. We labelled each trajectory $\tau$ in $\mathcal{D}$ with an LTL description $\varphi$ (based on the propositional labels for $\tau$), then trained the policy to maximize the log-likelihood of actions in $\tau$, conditioned on the history $h_t$ and $\varphi$. For each downstream task in Table 1, we prompted the policy with an LTL formula that aligns with that task.

We also trained bespoke models with advance knowledge of the downstream RM tasks. **Bespoke Reward Model** directly predicts rewards, optimal values, and terminations for all downstream tasks simultaneously. We labelled each trajectory in $\mathcal{D}$ with ground-truth rewards and terminations for each task based on the propositional labels, then trained the model to directly predict these quantities given the history $h_t$. Value estimates were trained via offline RL in a similar manner to

Table 1: List of RM tasks. For each, we report the mean ($\mu_{\mathcal{D}}$) and max ($\max_{\mathcal{D}}$) undiscounted return over trajectories in $\mathcal{D}$, along with the max achievable expected return of any policy (Max; if unknown, we report the highest average return observed in our experiments). Some tasks involve behaviours that are rarely or never observed in $\mathcal{D}$.

| Task | Description | Return | | |
|---|---|---|---|---|
| *GeoGrid* | | $\mu_{\mathcal{D}}$ | $\max_{\mathcal{D}}$ | Max |
| Sequence | Go to a red $\triangle$, then a green $\triangle$, then a blue $\triangle$. | 0.04 | 1 | 1 |
| Loop | Repeatedly go to a red $\triangle$, then a green $\triangle$, then a blue $\triangle$. | 0.04 | 3 | 5.36 |
| Logic | Go to all six objects, but always go to red objects before blue objects, and blue objects before green objects. | 0.00 | 1 | 1 |
| Safety | Go to a red object, then a blue object, then a green object, but always avoid $\triangle$. | −0.84 | 1 | 1 |
| *DrawerWorld* | | | | |
| Hold-Red-Box | Lift and hold the red box as long as possible. | 41.7 | 736 | 1538 |
| Pickup-Each-Box | Pick up the red box, then the blue box, then the green box. | 0 | 0 | 1 |
| Show-Green-Box | Reveal the green box if it's in a closed drawer, then lift it. | 0.22 | 1 | 1 |

GCR. Finally, we trained a policy via RL while using the learned value function for potential-based reward shaping. **Bespoke Behaviour Cloning (BC)** is a neural network policy that directly imitates successful trajectories in $\mathcal{D}$ for every downstream task. Due to the limited number of reward-worthy trajectories in $\mathcal{D}$, we considered any trajectory achieving positive return on that task as successful.

We also compared various reward shaping schemes for GCR. **No RS** directly uses RM rewards without any reward shaping (i.e. Algorithm 1). **High-Level RS**, inspired by Camacho et al. [34], uses a potential function that only considers the current RM state, but not the current MDP state.

## 7.3 Results

We ran each method's training pipeline five times and report the average final performances in Table 2. Performance was measured by undiscounted return (averaged over 100 evaluation episodes for GeoGrid or 20 for DrawerWorld), where rewards are with respect to the ground-truth labelling function $\mathcal{L}^*$. In Appendix B.4, we report RL learning curves, both with respect to the agent's own reward model (without shaping rewards) and ground-truth rewards under $\mathcal{L}^*$.

*GCR consistently solves novel tasks, even when no successful demonstrations exist in $\mathcal{D}$.* On Loop, Hold-Red-Box and Pickup-Each-Box, it significantly outperforms even the best trajectories in $\mathcal{D}$. We attribute this success to GCR's ability to learn transferable knowledge from $\mathcal{D}$ and apply it towards novel task compositions, while fine-tuning behaviours with (self-supervised) RL.

*All non-compositional baselines fail to reliably solve any task.* Results show that LTL-BC, Bespoke Reward Model, and Bespoke BC do not fare well with limited pretraining trajectories. Learning curves show that the Bespoke Reward Model assigns near-zero rewards to most trajectories in GeoGrid, likely due to the rarity of positive demonstrations in $\mathcal{D}$. In DrawerWorld, it produces misaligned rewards, leading to reward hacking (evidenced by high returns under the learned reward model but low returns under the ground-truth $\mathcal{L}^*$).

*Reward shaping enables long-horizon RL.* Our reward shaping strategy yields modest improvements in GeoGrid, but is critical to success in DrawerWorld, where behaviours like opening a drawer and picking up a box are nearly impossible to discover from random exploration alone.

We conclude the following. GCR faithfully elicits behaviours from RM specifications, outperforming non-compositional approaches (**RQ1**). Moreover, GCR compositionally generalizes to OOD behaviours beyond those observed in $\mathcal{D}$ (**RQ2**). Finally, our compositional reward shaping strategy for GCR enables RL in long-horizon settings involving propositional sparsity (**RQ3**).

## 7.4 Extending Ground-Compose-Reinforce with a Natural Language Interface

While natural language (NL) is often argued to have compositional properties [75], exploiting this compositionality in agentic language models (e.g. vision-language-action models) remains an open challenge. In Appendix C, we show that our GeoGrid RMs can be *autoformalized* directly from an

Table 2: Comparison of methods for eliciting behaviours from high-level task specifications. We report performance (undiscounted return with respect to ground-truth rewards) averaged over 5 runs with standard error.

| Task | GCR (Ours) | LTL-BC | Bespoke Reward Model | Bespoke BC | GCR (Ours) No RS | GCR (Ours) High-Level RS |
|---|---|---|---|---|---|---|
| *GeoGrid* | | | | | | |
| Sequence | $\mathbf{1.00} \pm 0.00$ | $0.04 \pm 0.01$ | $0 \pm 0$ | $0.05 \pm 0.01$ | $0.94 \pm 0.03$ | $\mathbf{1.00} \pm 0.00$ |
| Loop | $\mathbf{5.36} \pm 0.08$ | $0.03 \pm 0.01$ | $0 \pm 0$ | $0.04 \pm 0.01$ | $4.68 \pm 0.05$ | $5.27 \pm 0.08$ |
| Logic | $\mathbf{0.94} \pm 0.01$ | $0 \pm 0$ | $0 \pm 0$ | $0 \pm 0$ | $0.00 \pm 0.00$ | $\mathbf{0.94} \pm 0.01$ |
| Safety | $\mathbf{1.00} \pm 0.00$ | $-0.84 \pm 0.01$ | $-0.14 \pm 0.01$ | $-0.85 \pm 0.01$ | $0.23 \pm 0.11$ | $0.97 \pm 0.01$ |
| *DrawerWorld* | | | | | | |
| Hold-Red-Box | $\mathbf{1538} \pm 130$ | $0 \pm 0$ | $0 \pm 0$ | $0 \pm 0$ | $0 \pm 0$ | $0 \pm 0$ |
| Pickup-Each-Box | $\mathbf{1.00} \pm 0.00$ | $0 \pm 0$ | $0 \pm 0$ | $0 \pm 0$ | $0 \pm 0$ | $0 \pm 0$ |
| Show-Green-Box | $\mathbf{0.61} \pm 0.06$ | $0 \pm 0$ | $0 \pm 0$ | $0 \pm 0$ | $0 \pm 0$ | $0 \pm 0$ |

NL reward function description using OpenAI's o3 model, *zero-shot*—i.e., without fine-tuning on trajectories or other forms of grounding in our specific environments. Thus, we posit that leveraging compositional representations like RMs can be an effective way of building NL-interfaced agents in settings with limited labelled trajectory data (i.e. where $|\mathcal{D}|$ is small).

## 8 Future Work and Limitations

**Extension to Other Compositional Representations:** In this work, we propose an end-to-end framework for grounding high-level specifications in behaviours that leverages the compositionality inherent in RMs. However, we believe our core insights apply to a wide range of compositional representations such as those that deal with objects [76] and relations [77].

**Extension to Other Problem Settings:** Grounding language is a prerequisite for a myriad of language-conditioned problem settings. We consider an "RL-in-the-loop" setting, but future works could extend our insights to zero-shot execution of language tasks [5, 40, 46], question answering [78, 79], and interactive task learning [80, 81].

**Reward Hacking:** Misalignment between an agent's interpretation of a task and human intent can lead to harmful consequences, particularly in RL [82]. The use of formal specifications like RMs, which are unambiguous over the propositional vocabulary, can partially mitigate this, but ambiguity in the propositions themselves remains a concern in Ground-Compose-Reinforce. Prior works suggest that RM structure can be exploited to improve decision making under such ambiguity [55, 56].

**Assumptions on $\mathcal{D}$:** We assume that trajectories in $\mathcal{D}$ are labelled with values for a fixed set of propositions. Future works could explore other representations of propositions (e.g. as text) as well as scalable labelling methods (e.g. crowdsourced annotations [69] or self-supervised learning [70]).

## 9 Conclusion

This work presents Ground-Compose-Reinforce, an end-to-end framework for training RL agents directly from Reward Machine specifications—without oracle reward or labelling functions. A key challenge that we address is *grounding* these high-level task specifications in executable behaviours, given an agent's perception and action capabilities. We find that exploiting compositional task structure is critical to faithfully capturing this grounding from limited data. Starting from only 350 labelled pretraining trajectories, we show that our technical approach scales to temporally extended manipulation tasks in Meta-World while generalizing out-of-distribution to behaviours that never appear in pretraining. Moreover, we show that in some cases, Reward Machines can be autoformalized directly from natural language reward function descriptions to expose this temporal task structure.

More broadly, we show that leveraging language *compositionality* presents a promising pathway to building language-driven agents *without* relying on massive language-labelled data. Future work could explore the extension of these ideas to large-scale agentic language models such as vision-language-action models.

## Acknowledgements

We thank Harris Chan for his insightful and valuable input throughout all stages of this project. We gratefully acknowledge funding from the Natural Sciences and Engineering Research Council of Canada (NSERC) and the Canada CIFAR AI Chairs Program. Resources used in preparing this research were provided, in part, by the Province of Ontario, the Government of Canada through CIFAR, and companies sponsoring the Vector Institute for Artificial Intelligence (`https://vectorinstitute.ai/partnerships/`). Finally, we thank the Schwartz Reisman Institute for Technology and Society for providing a rich multi-disciplinary research environment.

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

# Supplementary Material for *Ground-Compose-Reinforce: Tasking Reinforcement Learning Agents through Formal Language*

In this supplementary material,

- we provide further details and analysis on Ground-Compose-Reinforce (Appendix A)
- we provide experimental details (Appendix B),
- we show how we autoformalize natural language descriptions of reward functions into our Reward Machines with LLMs (Appendix C),
- and we discuss the potential societal impact of this work (Appendix D).

## A    Supplementary Details on Ground-Compose-Reinforce

In this section, we

- discuss when Ground-Compose-Reinforce is likely to work well in practice (Appendix A.1)
- elaborate on Section 6.1's description of how we approximate the optimal value function $V_{\mathcal{R}}^*(s, u)$,
    - first under the assumption that the RM does not contain self-loop transitions with non-zero rewards (Appendix A.2),
    - and then relaxing that assumption (Appendix A.3);
- discuss approximation errors that can occur (Appendix A.4);
- describe how we train with offline RL the *primitive value functions* used in the approximation (Appendix A.5);
- show how we use the approximate value functions for potential-based reward shaping (Appendix A.6).

### A.1    Analysis of Advantages and Assumptions

For Ground-Compose-Reinforce to work well in practice, we assume that a faithful grounding of propositions can be learned during the pretraining phase from $\mathcal{D}$ (i.e. $\hat{\mathcal{L}}(s) \approx \mathcal{L}^*(s)$). If this is the case, the rewards generated by the core framework (Algorithm 1) will faithfully capture tasks for any RM over propositions $\mathcal{AP}$, by design. To achieve this, $\mathcal{D}$ should provide sufficient coverage of the state space $\mathcal{S}$—but critically, it does not require sufficient coverage of the space of possible trajectories to generalize. Hence, Ground-Compose-Reinforce is able to reliably elicit desirable trajectories that are significantly out-of-distribution with respect to $\mathcal{D}$.

Unlike many imitation-learning-based methods, Ground-Compose-Reinforce also does not rely on a high concentration of expert demonstrations in $\mathcal{D}$ (as evidenced by the fact that our agent reliably solves tasks, even when $\mathcal{D}$ contains only random-action trajectories). We can attribute this to the RL phase, where the agent fine-tunes its policy using its self-generated learning signal.

### A.2    Approximating $V_{\mathcal{R}}^*(s, u)$ (No Rewarding RM Self-Transitions)

To estimate $V_{\mathcal{R}}^*(s, u)$, we evaluate each outgoing transition $\langle u, u', \varphi, r \rangle$ from $u$ by combining the subtask value $V_{\diamond \varphi}^*(s)$ with $v_{\mathcal{R}}^*(u')$. While Camacho et al. [34] treat RM transitions as singular actions, we treat RM transitions as temporally extended *options* [83] that take a variable number of steps to execute. Specifically, for a transition $\langle u, u', \varphi, r \rangle$, we make the following assumptions about the corresponding option:

(1) It can be initiated if and only if the current RM state is $u$.

(2) It optimizes for the subtask $\diamond \varphi$ (i.e. satisfying $\varphi$ quickly and with high probability).

(3) It either terminates when $\varphi$ is satisfied after a variable length of time $K$, or it never terminates.

(4) The option will never result in a different transition in the RM than the one in question.

**Algorithm 2** Value Iteration over RM States (Modified from Camacho et al. [34])

**Input:** RM $\mathcal{R} = \langle \mathcal{U}, u_0, \mathcal{F}, \mathcal{AP}, \delta_u, \delta_r \rangle$, discount factor $\gamma_{\text{RM}}$
1: Initialize $v_{\mathcal{R}}^*(u) \leftarrow 0, \forall u \in \mathcal{U}$
2: **while** not converged **do**
3:    **for** $u$ in $\mathcal{U}$ **do**
4:       $v_{\mathcal{R}}^*(u) \leftarrow \max_{\langle u', \varphi, r \rangle \in \delta(u)} \gamma_{\text{RM}}(r + v_{\mathcal{R}}^*(u'))$
5: **return** $v_{\mathcal{R}}^*$

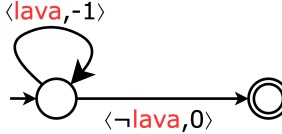

Figure 5: An RM that produces a reward of $-1$ for each timestep the agent spends in lava, until it exits the lava.

To estimate $v_{\mathcal{R}}^*(u)$, we run a variant of Value Iteration, modified from Camacho et al. [34] to reflect that rewards are garnered *after* the option completes (Algorithm 2). Note that Algorithm 2 should be run with a high-level discount factor $\gamma_{\text{RM}} \ll \gamma$ as it treats RM transitions as singular actions while in the actual RM-MDP, an RM transition may require many low-level steps to achieve.

We estimate the return for an RM transition $\langle u, u', \varphi, r \rangle$, given that the agent is in MDP state $s$ and RM state $u$, as follows. First, we model the return for *immediately* achieving the transition as $r + \gamma v_{\mathcal{R}}^*(u')$ — reward $r$ is immediately received for achieving the RM transition and $v_{\mathcal{R}}^*(u')$ estimates the future discounted return for being in RM state $u'$. If instead the RM transition is achieved $k$ steps in the future, we estimate the discounted return as $\gamma^k(r + \gamma v_{\mathcal{R}}^*(u'))$. Treating $\langle u, u', \varphi, r \rangle$ as an option that terminates when the RM transition is achieved, define $K$ to be the (random variable) number of steps it takes for this option to terminate when initiated from MDP state $s$ and RM state $u$ (where $K$ is $\infty$ if the RM transition is never achieved). The expected return of initiating the option is:

$$
\begin{aligned}
\mathbb{E}_{k \sim K}\left[ \gamma^k(r + \gamma v_{\mathcal{R}}^*(u')) \right] &= (\mathbb{E}_{k \sim K}[\gamma^k])(r + \gamma v_{\mathcal{R}}^*(u')) \\
&= V_{\Diamond \varphi}^*(s) \cdot (r + \gamma v_{\mathcal{R}}^*(u'))
\end{aligned}
$$

Recalling that the option is assumed to optimize for the subtask $\Diamond \varphi$, the final step results from the fact that $\mathbb{E}_{k \sim K}[\gamma^k]$ is precisely the optimal value for $\Diamond \varphi$ from state $s$. We finally estimate the optimal value function for RM task $\mathcal{R}$ by maximizing over the choice of option:

$$
V_{\mathcal{R}}^*(s, u) \approx \max_{\langle u', \varphi, r \rangle \in \delta(u)} \left[ V_{\Diamond \varphi}^*(s)(r + \gamma v_{\mathcal{R}}^*(u')) \right]
$$

### A.3   Approximating $V_{\mathcal{R}}^*(s, u)$ with Rewarding Self-Loop Transitions

To handle RMs that contain self-loop transitions with non-zero rewards, we require a few modifications. Intuitively, an option corresponding to a transition $\langle u, u', \varphi, r \rangle$ where $u \neq u'$ might receive a reward $r'$ at any timestep it is active if there exists another transition $\langle u, u, \varphi', r' \rangle$. As a simple example, consider the RM in Figure 5. The RM describes a task where the agent needs to exit the lava as quickly as possible, and receives a reward of $-1$ for each step until it does so. When estimating the expected return of the edge labelled $\langle \neg lava, 0 \rangle$, we need to consider the accumulation of the $-1$ reward at each timestep.

This can be hard to handle in general without more information, particularly when multiple self-loop edges exist for the same RM state. For simplicity, we assume that only non-self-loop transitions are treated as options (for the purposes of estimating optimal values), while *all* self-loop transitions in the current state $u$ are available at each step while an option is being executed. Thus, if we define

$r_{u,u}$ as the *maximum* reward for any self-loop transition in RM state $u$, we can modify our expression for the expected return of an option corresponding to transition $\langle u, u', \varphi, r \rangle$ in state $\langle s, u \rangle$ as follows:

$$\mathbb{E}_{k \sim K, s'} \left[ r_{u,u} + \gamma r_{u,u} + \ldots + \gamma^{k-1} r_{u,u} + \gamma^k (r + \gamma V_{\mathcal{R}}^*(s', u')) \right]$$

$$= \mathbb{E}_{k \sim K, s'} \left[ r_{u,u} \left( \frac{1 - \gamma^k}{1 - \gamma} \right) + \gamma^k (r + \gamma V_{\mathcal{R}}^*(s', u')) \right]$$

$$= \mathbb{E}_{s'} \left[ r_{u,u} \left( \frac{1 - V_{\diamond \varphi}^*(s)}{1 - \gamma} \right) + V_{\diamond \varphi}^*(s)(r + \gamma V_{\mathcal{R}}^*(s', u')) \right]$$

$$\approx r_{u,u} \left( \frac{1 - V_{\diamond \varphi}^*(s)}{1 - \gamma} \right) + V_{\diamond \varphi}^*(s)(r + \gamma v_{\mathcal{R}}^*(u'))$$

For the previous lava example, the expected return for the option that corresponds to the transition labelled $\langle \neg lava, 0 \rangle$ now correctly reflects a reward of $-1$ obtained for each step until the option terminates by the agent exiting the lava. We modify our approximation of $V_{\mathcal{R}}^*(s, u)$ as follows:

$$V_{\mathcal{R}}^*(s, u) \approx \max_{\langle u', \varphi, r \rangle \in \delta(u)} \left[ r_{u,u} \left( \frac{1 - V_{\diamond \varphi}^*(s)}{1 - \gamma} \right) + V_{\diamond \varphi}^*(s)(r + \gamma v_{\mathcal{R}}^*(u')) \right] \tag{4}$$

We also modify the Value Iteration algorithm to consider self-loops with non-zero rewards (Algorithm 3). Recall that the Value Iteration algorithm represents the RM as a high-level MDP where RM transitions are treated as singular actions under a discount factor $\gamma_{\text{RM}}$. However, in the actual MDP (with discount factor $\gamma$), an RM transition may take several steps to be achieved. One way of viewing the high-level MDP is that it makes a simplifying assumption that all RM transitions take some random variable number of steps $k \sim K$ to complete (where $K$ is the same for all RM transitions) and $\gamma_{\text{RM}} = \mathbb{E}_{k \sim K}[\gamma^k]$. In other words, $\gamma_{\text{RM}}$ represents the expected amount of discounting upon the RM transition's completion.

Now, consider an RM transition from state $u$ to $u'$ that receives reward $r$, but that self-loops in state $u$ for $k \sim K$ steps (receiving reward $r_{u,u}$ on each step) before the transition completes. Under the aforementioned assumptions, the expected discounted return of taking this RM transition, including the self-loop rewards, is:

$$\mathbb{E}_{k \sim K} \left[ \left( \sum_{t=0}^{k-1} \gamma^t r_{u,u} \right) + \gamma^k (r + v_{\mathcal{R}}^*(u')) \right] \tag{5}$$

$$= \mathbb{E}_{k \sim K} \left[ r_{u,u} \left( \frac{1 - \gamma^k}{1 - \gamma} \right) + \gamma^k (r + v_{\mathcal{R}}^*(u')) \right] \tag{6}$$

$$= r_{u,u} \left( \frac{1 - \gamma_{\text{RM}}}{1 - \gamma} \right) + \gamma_{\text{RM}} (r + v_{\mathcal{R}}^*(u')) \tag{7}$$

---

**Algorithm 3** Value Iteration over RM States (modified for self-loop transitions)

---

**Input:** RM $\mathcal{R} = \langle \mathcal{U}, u_0, \mathcal{F}, \mathcal{AP}, \delta_u, \delta_r \rangle$, MDP discount factor $\gamma$, high-level discount factor $\gamma_{\text{RM}}$
1: Initialize $v_{\mathcal{R}}^*(u) \leftarrow 0, \forall u \in \mathcal{U}$
2: Extract maximum self-loop rewards $r_{u,u}, \forall u \in \mathcal{U}$ from $\delta_r$
3: **while** not converged **do**
4:    **for** $u$ in $\mathcal{U}$ **do**
5:       $v_{\mathcal{R}}^*(u) \leftarrow \max_{\langle u', \varphi, r \rangle \in \delta(u)} \left( r_{u,u} \left( \frac{1 - \gamma_{\text{RM}}}{1 - \gamma} \right) + \gamma_{\text{RM}} (r + v_{\mathcal{R}}^*(u')) \right)$
6: **return** $v_{\mathcal{R}}^*$

---

### A.4 Sources of Approximation Errors

We now discuss how errors can occur when estimating $V_{\mathcal{R}}^*(s, u)$ from PVFs.

First, Approximation 4 clearly introduces approximation error from assuming the largest reward among self-loop transitions $r_{u,u}$ will be garnered until an outgoing transition is reached, and from using a bootstrapped value estimate of the next state $v_{\mathcal{R}}^*(u') \approx V_{\mathcal{R}}^*(s', u')$. Another less obvious

source of error is that the expected return for each outgoing transition $\langle u, u', \varphi, r \rangle$ is estimated by assuming that an optimal policy for reaching $\varphi$ will be followed. However, this ignores the possibility of some *other* transition from $u$ occurring before the intended transition. In general, it may not be possible to satisfy $\varphi$ while achieving value $V^*_{\diamond\varphi}(s)$ in the subtask $\diamond\varphi$ while also avoiding all other transitions from RM state $u$ that are not $\langle u, u', \varphi, r \rangle$.

Estimating the OVF for a disjunction of formulas via Approximation 2 always *underestimates* the true value. We prove this as follows. Suppose $\varphi = \xi_1 \vee \ldots \vee \xi_k$ and recall that the approximation of $V^*_{\diamond\varphi}(s)$ is $\max_{i=1,\ldots,k} V^*_{\diamond\xi_i}(s)$. Observe that for each $i$, $V^*_{\diamond\xi_i}(s) \leq V^*_{\diamond\varphi}(s)$, since for every trajectory $\tau$, if $\xi_i$ is satisfied at timestep $T$ in $\tau$, then $\varphi$ must also be satisfied at timestep $T$ (or earlier) in $\tau$. Thus, $\max_{i=1,\ldots,k} V^*_{\diamond\xi_i}(s) \leq V^*_{\diamond\varphi}(s)$. The reason this bound is not tight is because there are situations where satisfying *one of* $\xi_1, \ldots, \xi_k$ is easier than satisfying any of $\xi_1, \ldots, \xi_k$ individually. For example, consider the task $\diamond(X \vee \neg X)$, which is always trivially solved on the first step. However, it is possible that $\diamond X$ and $\diamond \neg X$ are both non-trivial tasks.

Lastly, estimating the OVF for a conjunction of formulas via Approximation 3 always *overestimates* the true value, by a similar line of reasoning as for disjunction. In general, knowing $V^*_{\diamond\xi_1}(s)$ and $V^*_{\diamond\xi_2}(s)$ does not provide enough information to estimate $V^*_{\diamond(\xi_1 \wedge \xi_2)}(s)$. For instance, it may be the case that $\xi_1, \xi_2$ are mutually exclusive and thus, $V^*_{\diamond(\xi_1 \wedge \xi_2)}$ is zero everywhere. However, whether or not $\xi_1, \xi_2$ are mutually exclusive cannot be inferred based only on $V^*_{\diamond\xi_1}(s)$ and $V^*_{\diamond\xi_2}(s)$. Nangue Tasse et al. [84] provide a discussion on this topic for a related setting.

### A.5 Training Primitive Value Functions

PVFs can be trained directly from the trajectory dataset $\mathcal{D}$ based on any offline RL approach. Algorithm 4 shows how to train the PVF for a single primitive task $\diamond x$ using a simple offline RL algorithm (Fitted Q-Iteration [85]). The approach can be easily adapted to negations $\diamond \neg x$ as well, and in practice, we simultaneously train all $2|\mathcal{AP}|$ possible PVFs in parallel as a single neural network.

---

**Algorithm 4** Learning PVF $V^*_{\diamond_x}(s)$ for $x \in \mathcal{AP}$ from Trajectory Data $\mathcal{D}$

---

**Input:** Dataset $\mathcal{D} = \left\{ \langle \tau^i, \omega^i \rangle \right\}_{i=1}^N$, discount factor $\gamma$, proposition $x \in \mathcal{AP}$
1: Initialize Q function $Q_\theta : \mathcal{S} \times \mathcal{A} \to \mathbb{R}$
2: Initialize value function $V_\psi : \mathcal{S} \to \mathbb{R}$
3: **while** not converged **do**
4:     Sample transition $\langle s, \omega, a, s', \omega' \rangle \sim \mathcal{D}$
5:     Update $\phi$ with SGD on $\text{BinaryCrossEntropy}(\mathcal{L}_\phi(s), \mathbb{1}[x \in \omega])$
6:     reward $\leftarrow \mathbb{1}\left[x \in \omega'\right]$, next_value $\leftarrow \max_{a' \in \mathcal{A}} Q_\theta(s', a')$, done $\leftarrow \mathbb{1}\left[x \in \omega'\right]$
7:     Update $\theta$ with SGD on $\left( Q_\theta(s, a) - \text{stop\_grad}(\text{reward} + \gamma * (1 - \text{done}) * \text{next\_value}) \right)^2$
8:     Update $\psi$ with SGD on $(V_\psi(s') - \text{next\_value})^2$
9: **return** $V_\psi$

---

### A.6 Potential-based Reward Shaping

We extend our core Ground-Compose-Reinforce algorithm with potential-based reward shaping by leveraging Approximations 1-3 and trained PVFs to predict $V^*_{\mathcal{R}}(s, u)$ for any MDP state $s$ and RM state $u$. This is shown in Algorithm 5 with changes from the core algorithm highlighted in red.

## B  Experimental Details

### B.1 Domain Descriptions

**Environments.** *GeoGrid* is an $8 \times 8$ image-based gridworld depicted in Figure 2 with six objects randomly positioned at the start of each episode. States are $8 \times 8 \times 6$-dimensional images that identify each cell's colour/shape and the agent's location, while propositions $\{R, G, B, \triangle, \bigcirc\}$ identify if the agent is at an object with that particular colour or shape. *DrawerWorld* is a MuJoCo environment adapted from Meta-World [17]. The agent controls a robotic gripper and can interact with two

---

**Algorithm 5** Ground-Compose-Reinforce for RMs with Potential-Based Reward Shaping

---

**Input:** MDP $\mathcal{M}$ without rewards, Propositional symbols $\mathcal{AP}$, Dataset $\mathcal{D}$ of labelled trajectories, RM task $\mathcal{R}$ over $\mathcal{AP}$, Shaping potential weighting coefficient $\lambda$
    *// Pretraining phase*
1: Train labelling function $\hat{\mathcal{L}}(s)$ on $\mathcal{D}$ using any binary classification method
2: Train PVFs $V^*_{\Diamond x}(s)$ and $V^*_{\Diamond \neg x}(s)$, $\forall x \in \mathcal{AP}$ on $\mathcal{D}$
    *// Behaviour elicitation phase*
3: Initialize policy $\pi_{\mathcal{R}}(a \mid s, u)$ arbitrarily
4: **for** each episode **do**
5:     Observe initial state $s$ in $\mathcal{M}$; set $u$ to the initial state of $\mathcal{R}$
6:     Estimate initial value $v \approx V^*_{\mathcal{R}}(s, u)$ using Approximations 1-3 and trained PVFs
7:     **while** $u$ is non-terminal **do**
8:         Sample action $a \sim \pi_{\mathcal{R}}(\cdot \mid s, u)$
9:         Execute $a$ in $\mathcal{M}$ and observe next state $s'$
10:       Compute truth assignment $\hat{\omega} \leftarrow \hat{\mathcal{L}}(s')$
11:       Update RM: $u' \leftarrow \delta_u(u, \hat{\omega})$, $r \leftarrow \delta_r(u, \hat{\omega})$
12:       Update value $v' \approx V^*_{\mathcal{R}}(s', u')$ using Approximations 1-3 and trained PVFs
13:       Update policy $\pi_{\mathcal{R}}$ with RL for transition $\langle s, u, a, r + \lambda(\gamma v' - v), s', u' \rangle$
14:       Set $s \leftarrow s'$, $u \leftarrow u'$, $v \leftarrow v'$

---

drawers (left and right) and three boxes (red, green, and blue). Observations are 78-dimensional vectors representing positions of objects and the gripper. Propositions identify whether a particular drawer is open, whether a particular block is picked up by the agent, and whether a particular block is currently inside a particular drawer.

**Datasets.** We carefully curated datasets $\mathcal{D}$ in each environment to support our analysis of compositional generalization (RQ2). In GeoGrid, $\mathcal{D}$ is comprised of 5000 trajectories of length 100 *generated under a random-action policy*. In DrawerWorld, $\mathcal{D}$ is comprised of 350 trajectories of varying length that we collected by manually controlling the robot in the simulator. To ensure sufficient state coverage in $\mathcal{D}$, drawers and boxes were initialized in a random configuration when collecting each trajectory. Behaviours that appear in the dataset include opening and closing drawers, picking up boxes, and moving boxes from one location to another, but *no trajectory involves direct interaction with more than one box*. A small number of trajectories involve incidental (but not prolonged) interaction with more than one box (e.g. bumping into one box while moving another). We intentionally include accidental behaviours in the DrawerWorld dataset such as failing to grip a box, dropping a box while attempting to move it, and opening a drawer beyond its limit. We also include behaviours not tied to downstream tasks such as placing a box on top of a drawer or throwing a box off the table.

**Tasks.** We designed a diverse set of RM tasks (Table 1). For tasks that involve achieving a (temporally extended) goal, the RM terminates and provides a reward of 1 upon doing so. The RM terminates with a reward of 0 if the goal becomes logically impossible (e.g. due to breaking a constraint), except in Safety, which terminates with a penalty of $-1$. Loop and Hold-Red-Box involve repeating some desired behaviour, and the RM yields a reward of 1 for each such repetition. For the precise encodings of tasks as RMs, please see the released code.

### B.2 Baseline Implementation Details

All approaches involve supervised training on $\mathcal{D}$, and Ground-Compose-Reinforce and Bespoke Reward Model additionally require an RL phase in the environment. Network architectures for supervised training on $\mathcal{D}$ are reported in Table 3. PPO network architectures are reported in Table 4. All policy networks (whether trained via RL or behaviour cloning) use GRUs to model temporal dependencies except for GCR, which uses RM transitions. The policy network outputs a probability distribution over actions (in DrawerWorld, the outputs of the network parameterize a Gaussian policy's mean and standard deviation). For methods relying on potential-based reward shaping, we computed shaping rewards without discounting the next potential; i.e., we issued the shaping reward as $\lambda(v' - v)$ rather than $\lambda(\gamma v' - v)$. Though this loses some theoretical convergence properties, we found it to significantly outperform the standard shaping reward in all cases.

Table 3: Network Architectures for Supervised Training on $\mathcal{D}$.

| | GCR | | LTL-BC | Bespoke Reward Model | Bespoke BC |
|---|---|---|---|---|---|
| Labelling Function | | PVFs | | | |
| *GeoGrid* | | | | | |
| `Conv2d(6,16,3,1,1)`
`ReLU`
`Conv2d(16,32,3,1,1)`
`ReLU`
`Flatten`
`Linear(2048,128)`
`ReLU`
`Linear(128,5)` | `Conv2d(6,32,3,1,1)`
`ReLU`
`Conv2d(32,32,3,1,1)`
`ReLU`
`Flatten`
`Linear(2048,256)`
`ReLU`
`Linear(256,10)` | | **Obs Encoder:**
`Conv2d(6,16,3,1,1)`
`ReLU`
`Conv2d(16,32,3,1,1)`
`ReLU`
`Flatten`
`Linear(2048,256)`

**LTL Encoder:**
`Transformer(d_model=64,`
` nhead=4,`
` dim_feedforward=128,`
` num_layers=2)`

**Policy:**
`GRU(256+64,256)`
`ReLU`
`GRU(256,256)`
`Linear(768,16)` | `Conv2d(6,16,3,1,1)`
`ReLU`
`Conv2d(16,32,3,1,1)`
`ReLU`
`Flatten`
`Linear(2048,256)`
`GRU(256,256)`
`ReLU`
`GRU(256,256)`
`Linear(768,4)` $\times$ 3 | `Conv2d(6,16,3,1,1)`
`ReLU`
`Conv2d(16,32,3,1,1)`
`ReLU`
`Flatten`
`Linear(2048,256)`
`GRU(256,256)`
`ReLU`
`GRU(256,256)`
`Linear(768,16)` |
| *DrawerWorld* | | | | | |
| `Linear(39,1600)`
`ReLU`
`Linear(1600,11)` | `Linear(39,1600)`
`ReLU`
`Linear(1600,400)`
`ReLU`
`Linear(400,22)` | | **Obs Encoder:**
`Linear(39,1600)`
`ReLU`
`Linear(1600,400)`

**LTL Encoder:**
`Transformer(d_model=64,`
` nhead=4,`
` dim_feedforward=128,`
` num_layers=2)`

**Policy:**
`GRU(464,256)`
`ReLU`
`GRU(256,256)`
`Linear(976,4)` | `Linear(39,1600)`
`ReLU`
`Linear(1600,400)`
`ReLU`
`GRU(400,256)`
`ReLU`
`GRU(256,256)`
`Linear(912,3)` $\times$ 3 | `Linear(39,1600)`
`ReLU`
`Linear(1600,400)`
`ReLU`
`GRU(400,256)`
`ReLU`
`GRU(256,256)`
`Linear(912,24)` |

**Ground-Compose-Reinforce.** GCR consists of the following neural networks: a labelling function network that outputs a single binary classification logit for each proposition in $\mathcal{AP}$, a PVF network that outputs an optimal value prediction for each literal in $\mathcal{AP}$, and a policy of the form $\pi(a_t|s_t, u_t)$ that conditions on the current RM state. The labelling function is trained via a binary cross entropy loss on $\mathcal{D}$, the PVFs are trained via offline RL on $\mathcal{D}$, and the policy is trained via RL supported by the labelling function and PVFs to provide learning signals.

In GeoGrid, PVFs were trained using Fitted Q-Iteration [85]. In DrawerWorld, PVFs were trained to directly predict Monte Carlo returns. We also considered a state-of-the-art offline RL method, MCQ [86], but it performed worse than Monte Carlo regression. We attribute this to the relatively small size of $\mathcal{D}$ compared to standard offline RL benchmarks.

**LTL-conditioned Behaviour Cloning.** This baseline models a neural network policy $\pi_\theta(a_t|h_t, \varphi)$, where the history $h_t$ is encoded by a GRU [74] and $\varphi$ (a goal represented directly in LTL) is encoded by a Transformer [87]. Only observations (and not actions) are encoded as part of the history. We trained $\pi_\theta$ by labelling each trajectory $\tau^i$ in $\mathcal{D}$ with an LTL formula $\varphi^i$ based on the sequence of propositional labels $\sigma^i$ in $\mathcal{D}$ and then minimized the behaviour cloning loss $\mathbb{E}_\mathcal{D}[-\log \pi_\theta(a_t|h_t, \varphi)]$. Finally, we evaluated the model on the downstream tasks by conditioning on the LTL formulas shown in Table 5.

To the best of our knowledge, there are no existing approaches that generate LTL descriptions based on *a single trajectory*. We instead used a custom approach based on common specification templates to generate diverse LTL descriptions. For each trajectory $\tau$ in $\mathcal{D}$, we randomly generated a single formula that is satisfied by $\tau$ for each of the following specification templates found in Table 2 of Menghi et al. [88]: *visit*, *sequenced visit*, *ordered visit*, *patrolling* (for this purpose, we consider an event to occur infinitely often if it occurs at least five times within the same trajectory in GeoGrid or

Table 4: Network Architectures for PPO.

| GeoGrid | | DrawerWorld | |
|---|---|---|---|
| **GCR** | **Bespoke Reward Model** | **GCR** | **Bespoke Reward Model** |
| **Encoder:**
`Conv2d(6,16,3,1,1)`
`ReLU`
`Conv2d(16,32,3,1,1)`
`ReLU`
`Flatten`

**Actor Head:**
`Linear(2048+`$\|\mathcal{U}\|$`,128)`
`ReLU`
`Linear(128,64)`
`ReLU`
`Linear(64,4)`

**Critic Head:**
`Linear(2048+`$\|\mathcal{U}\|$`,128)`
`ReLU`
`Linear(128,64)`
`ReLU`
`Linear(64,1)` | **Encoder:**
`Conv2d(6,16,3,1,1)`
`ReLU`
`Conv2d(16,32,3,1,1)`
`ReLU`
`Flatten`
`GRU(2048,128)`

**Actor Head:**
`Linear(128,128)`
`ReLU`
`Linear(128,64)`
`ReLU`
`Linear(64,4)`

**Critic Head:**
`Linear(128,128)`
`ReLU`
`Linear(128,64)`
`ReLU`
`Linear(64,1)` | **Actor:**
`Linear(78+`$\|\mathcal{U}\|$`,512)`
`ReLU`
`Linear(512,512)`
`ReLU`
`Linear(512,512)`
`ReLU`
`Linear(512,8)`

**Critic:**
`Linear(78+`$\|\mathcal{U}\|$`,512)`
`ReLU`
`Linear(512,512)`
`ReLU`
`Linear(512,512)`
`ReLU`
`Linear(512,1)` | **Encoder:**
`Linear(78,512)`
`ReLU`
`Linear(512,512)`
`GRU(512,512)`

**Actor Head:**
`Linear(512,512)`
`ReLU`
`Linear(512,8)`

**Critic Head:**
`Linear(512,512)`
`ReLU`
`Linear(512,1)` |

200 times within the same trajectory in DrawerWorld) and *global avoidance*. These templates were chosen since they correspond to LTL properties that are relatively simple to automatically mine from a given trajectory. We then labelled each trajectory $\tau$ with a randomly chosen LTL formula from among this set.

**Bespoke Reward Model.** This baseline is a single neural network that directly predicts the reward, termination, and optimal value function for each of the downstream tasks. The neural network consists of an observation encoder, followed by two GRU layers (to encode the history of observations), followed by three linear output heads to predict rewards, optimal values, and terminations, respectively, for all downstream tasks simultaneously for that domain. To generate target rewards and terminations for a trajectory $\tau$ in $\mathcal{D}$, we evaluated the RM of each downstream task based on the sequence of propositional labels $\sigma^i$. The optimal value estimates were trained using offline RL in a similar manner as the PVFs. Finally, a policy was obtained using RL on the rewards and terminations, while the optimal values were used for potential-based reward shaping, similar to the shaped version of GCR.

**Bespoke Behaviour Cloning.** This baseline is similar to LTL-BC, except it does not condition on an LTL task—instead, it simultaneously outputs actions for each of the possible downstream tasks. To evaluate the policy on a specific downstream task, only the output for that task is considered. We trained the policy via behaviour cloning on any trajectory that achieves positive return on a particular downstream task due to the limited number of successful demonstrations.

## B.3 Experimental Setup and Hyperparameter Details

**Details: Supervised Training on $\mathcal{D}$.** All experiments were run on a compute cluster. Supervised training on $\mathcal{D}$ required a single GPU and CPU, minimal memory resources (24GB of RAM or less) and no more than 30 minutes to train any method to 100 epochs. We tuned hyperparameters via a line search over batch size, learning rate, L1 regularization coefficient, and epochs (in that order) using a held-out 10% of the trajectories in D, and the final hyperparameters are reported in Table 6. Final models were retrained on the full data with the tuned hyperparameters. We note that the batch size hyperparameter should be interpreted differently for methods requiring a GRU. For GCR, it refers to the number of transitions sampled from $\mathcal{D}$. For LTL-BC, Bespoke Reward Model and Bespoke Behaviour Cloning, it refers to the number of full length *trajectories* sampled from $\mathcal{D}$. This is because it is necessary to keep transitions in a trajectory in the correct order to train the GRU.

Table 5: LTL formulas used to evaluate LTL-BC.

| Task | LTL Formula |
|------|-------------|
| *GeoGrid* (propositions are named r,g,b,c,t instead of R, G, B, $\triangle$, $\bigcirc$ to avoid confusion with LTL operators) | |
| Sequence | $\Diamond((r \wedge t) \wedge \Diamond((g \wedge t) \wedge \Diamond(b \wedge t)))$ |
| Loop | $\Box\Diamond((r \wedge t) \wedge \Diamond((g \wedge t) \wedge \Diamond(b \wedge t)))$ |
| Logic | $\Diamond(r \wedge t) \wedge \Diamond(g \wedge t) \wedge \Diamond(b \wedge t) \wedge \Diamond(r \wedge c) \wedge \Diamond(g \wedge c) \wedge \Diamond(b \wedge c) \wedge (\neg b\,\mathcal{U}\,r) \wedge (\neg g\,\mathcal{U}\,b)$ |
| Safety | $\Diamond((r \wedge t) \wedge \Diamond((g \wedge t) \wedge \Diamond(b \wedge t))) \wedge \Box\neg t$ |
| *DrawerWorld* | |
| Hold-Red-Box | $\Box\Diamond\text{RedBoxLifted}$ |
| Pickup-Each-Box | $\Diamond(\text{RedBoxLifted} \wedge \Diamond(\text{BlueBoxLifted} \wedge \Diamond\text{GreenBoxLifted}))$ |
| Show-Green-Box | $\neg[\neg(\text{GreenBoxInDrawer1} \wedge \Diamond(\text{Drawer1Open}\,\mathcal{U}\,\text{GreenBoxLifted}))$ |
| | $\wedge\neg(\text{GreenBoxInDrawer2} \wedge \Diamond(\text{Drawer2Open}\,\mathcal{U}\,\text{GreenBoxLifted}))$ |
| | $\wedge\neg(\neg\text{GreenBoxInDrawer1} \wedge \neg\text{GreenBoxinDrawer2} \wedge \Diamond\text{GreenBoxLifted})]$ |

Table 6: Hyperparameters for Supervised Training on $\mathcal{D}$.

| Hyperparameter | GCR | | LTL-BC | Bespoke Reward Model | Bespoke BC |
|----------------|-----|-----|--------|----------------------|------------|
| | Labelling Function | PVFs | | | |
| *GeoGrid* | | | | | |
| Batch size | 256 | 1024 | 100 | 100 | 100 |
| Learning rate | 3e-4 | 3e-4 | 1e-4 | 3e-3 | 3e-4 |
| L1 loss | 1e-5 | 0 | 0 | 0 | 0 |
| Epochs | 10 | 100 | 9 | 100 | 11 |
| Discount factor | n.a. | 0.97 | n.a. | 0.97 | n.a. |
| *DrawerWorld* | | | | | |
| Batch size | 256 | 256 | 50 | 50 | 50 |
| Learning rate | 3e-4 | 3e-4 | 1e-3 | 1e-4 | 1e-4 |
| L1 loss | 1e-5 | 1e-5 | 0 | 0 | 1e-9 |
| Epochs | 100 | 100 | 87 | 11 | 4 |
| Discount factor | n.a. | 0.9975 | n.a. | 0.9975 | n.a. |

**Details: RL Training.** All experiments were run on a compute cluster. Each RL run used a single GPU, 16 CPUs, and 48GB of RAM. For GCR, runs took up to 6 hours on GeoGrid (to train to 15M frames) and 16 hours on DrawerWorld (to train to 20M frames). RL training with the Bespoke Reward Model took longer due to GRUs—up to 12 hours on GeoGrid (to train to 15M frames) and 18 hours on DrawerWorld (to train to 20M frames). For RL training, we used the implementation of PPO at `https://github.com/lcswillems/torch-ac` with the hyperparameters in Table 7. The total number of environment steps each method was trained on was 2.5M for Sequence, 4M for Loop, 10M for Safety, 20M for Show-Green-Box, and 15M for all others.

## B.4 Learning Curves

We report RL learning curves for each method and task in Figures 6 and 7. GCR and its variants have an internal reward model that is better aligned with the ground truth compared to Bespoke Reward Model. In GeoGrid, GCR's reward model is near perfect, and in all cases, optimizing its internal rewards improves ground truth performance as well. Bespoke Reward Model almost always predicts near-zero rewards on GeoGrid tasks (except Safety), likely since there are few examples of positive demonstrations in $\mathcal{D}$. On DrawerWorld, Bespoke Reward-Model is highly misaligned, predicting large rewards but garnering near-zero return based on the ground truth.

In terms of sample efficiency, we observe that GCR with our reward shaping strategy outperforms all baselines. The difference is marginal in GeoGrid, where exploration is less of an issue, but in DrawerWorld, all other reward shaping approaches fail.

Table 7: RL Training Hyperparameters.

| Hyperparameter | Value for all methods |
|---|---|
| *GeoGrid* | |
| Number of parallel environments | 16 |
| Frames per update per process | 1000 |
| Learning rate | 3e-4 |
| Discount factor ($\gamma$) | 0.97 |
| GAE parameter ($\lambda$) | 0.95 |
| Clip range | 0.2 |
| Entropy coefficient | 1e-4 |
| Value loss coefficient | 0.5 |
| Number of epochs per update | 4 |
| Minibatch size | 4000 |
| High-level Discount Factor ($\gamma_{\mathrm{RM}}$) | $0.97^{10}$ |
| Shaping potential weighting coefficient ($\lambda$) | 1 |
| *DrawerWorld* | |
| Number of parallel environments | 16 |
| Frames per update per process | 4000 |
| Learning rate | 3e-4 |
| Discount factor ($\gamma$) | 0.99 |
| GAE parameter ($\lambda$) | 0.99 |
| Clip range | 0.2 |
| Entropy coefficient | 0.01 in Pickup-Each-Box, otherwise 0.03 |
| Value loss coefficient | 0.5 |
| Number of epochs per update | 10 |
| Minibatch size | 8000 |
| High-level Discount Factor ($\gamma_{\mathrm{RM}}$) | $0.9975^{400}$ |
| Shaping potential weighting coefficient ($\lambda$) | 0.1 in Hold-Red-Box, otherwise 1 |

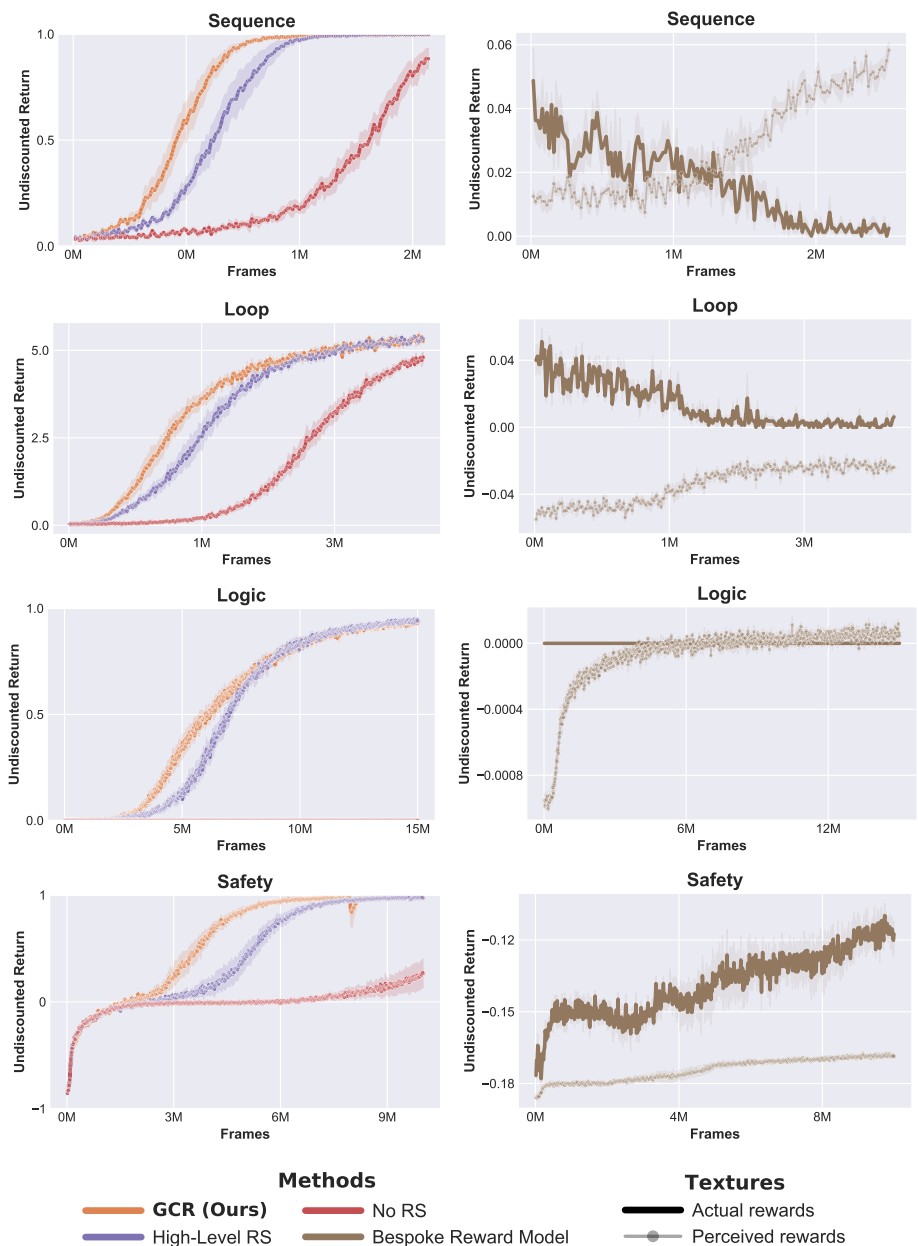

Figure 6: RL learning curves for GeoGrid, showing returns under the agent's own reward model ("perceived rewards") and under the ground-truth $\mathcal{L}^*$ ("actual rewards"). Perceived rewards are reported without shaped rewards, and shaded regions show standard error. Approaches based on Ground-Compose-Reinforce (including No RS and High-Level RS) generate rewards that are closely aligned with the ground truth and lead to an effective final policy, while Bespoke Reward Model produces near-zero rewards in all cases and makes little progress.

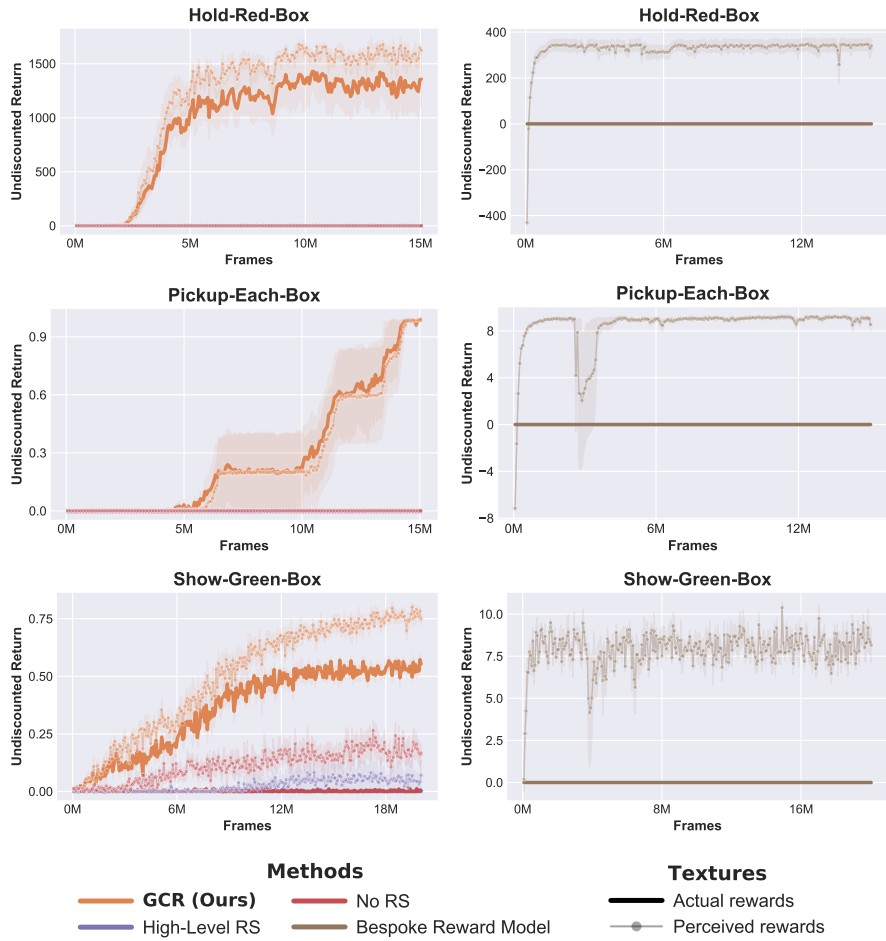

Figure 7: RL learning curves for DrawerWorld, showing returns under the agent's own reward model ("perceived rewards") and under the ground-truth $\mathcal{L}^*$ ("actual rewards"). Perceived rewards are reported without shaped rewards, and shaded regions show standard error. When evaluated under ground-truth rewards, Ground-Compose-Reinforce with our reward shaping strategy learns strong policies in all cases, while alternative approaches make no progress. Notably, Bespoke Reward Model results in a final policy with high *perceived rewards*, but poor actual performance.

## C   Autoformalizing Natural Language to Reward Machines

While several works have been dedicated to the autoformalization of natural language instructions into formal languages such as LTL [36–39], we show that a modern LLM can sometimes perform this task zero-shot for RMs, without having specifically been trained on it (to the best of our knowledge). This allows us to directly task the RL agent in our framework through natural language, then autoformalize the task description into an RM.

We tested OpenAI's ChatGPT-4o and o3 models as the autoformalizer and prompted it with a description of the autoformalization task (including the output format), a text description of the environment, a list of propositions and associated text descriptions, and a text description of the desired reward function (Listing 1). For each of the four GeoGrid tasks in Table 1, we ran each autoformalizer five times for consistency. We manually evaluated each outputted RM based on whether it yielded a reward function that exactly matched the textual description, with the success rate reported in Table 8.

ChatGPT-4o correctly produced RMs for all tasks, except for Logic, which requires a complex RM (our solution involved 10 states and 18 transitions). However, we note that ChatGPT-4o was nearly correct for all five trials for Logic—each of its outputted RMs deviated by a single transition that changed the behaviour of the resultant reward function. o3 outputted correct RMs on all tasks. Notably, the outputted RMs were identical in structure to the intended RMs we manually constructed in all cases (with the only differences being in the naming of RM states and the representation of equivalent logical formulas).

```
You are given a list of propositional symbols and their descriptions,
    an environment description, and a description of a desired reward
    function in English. Your job is to construct a Reward Machine
    representing this reward function.

Reward Machine states should be numbered 0, 1, 2, 3, ..., with 1
    always being the initial state, and 0 always being the terminal
    state. Transitions should be represented as a tuple (i, j, \varphi
    , r), where i and j are the start and end Reward Machine states of
     the transition, respectively, \varphi is a logical formula over
    the set of propositional symbols (use "!" to represent "not", "&"
    to represent "and", and "|" to represent "or", and write the
    formula in disjunctive normal form), and r is the reward for the
    transition. Your output should be the transitions in the Reward
    Machine, one per line, in the tuple form shown above, e.g. (0, 1,
    !X&Y, 0.1), with no other punctuation. For brevity, do not list
    self-loop transitions that provide 0 reward in the output.

Environment Description: The environment is a gridworld, where some
    squares have objects. Each object has a single colour (red, green,
     or blue) and a single shape (circle, or triangle).

Propositions:
- red: The agent's current cell has a red object.
- blue: The agent's current cell has a blue object.
- green: The agent's current cell has a green object.
- triangle: The agent's current cell has a triangle.
- circle: The agent's current cell has a circle.

Task: <TASK DESCRIPTION>
```

Listing 1: Reward Machine Autoformalization Prompt

## D   Societal Impact

**Data-efficient learning lowers environmental cost.** Ground-Compose-Reinforce (GCR) achieves strong generalization from a relatively small, task-agnostic trajectory dataset. Because it avoids the

Table 8: Success rate for ChatGPT-4o and o3 when autoformalizing Reward Machines from a natural language description of the desired reward function.

| Task | Description | GPT-4o Success Rate | o3 Success Rate |
|---|---|---|---|
| Sequence | Give a reward of 1 and terminate the episode when a red triangle, a green triangle, and a blue triangle have been reached, in that order. Only give the reward of 1 when the final step has been completed. Give 0 reward and never terminate the episode otherwise. | 100% | 100% |
| Loop | Give a reward of 1 when a red triangle, a green triangle, and a blue triangle have been reached, in that order. Only give the reward of 1 when the final step has been completed. After completing this sequence, the agent may repeat all steps of the sequence to receive the reward again, as many times as it wishes. The episode never terminates. | 100% | 100% |
| Logic | Give a reward of 1 and terminate the episode as soon as a red triangle, red circle, green triangle, green circle, blue triangle, and blue circle have all been reached at some point. However, blue objects should not be visited until both red objects are visited, and green objects should not be visited until both blue objects are visited. Circles and triangles of the same colour can be reached in either order. If any objects are reached out of order, immediately terminate the episode with a reward of 0. | 0% | 100% |
| Safety | Give a reward of 1 when a red object, a green object, and a blue object have been reached, in that order. Only give the reward of 1 when the final step has been completed. However, always avoid squares with triangles—if this is violated, immediately terminate the episode with a reward of -1. | 100% | 100% |

need for internet-scale demonstrations, the total compute and data collection burden is substantially reduced, which in turn diminishes the carbon footprint of training and retraining large decision-making systems.

**Transparent, verifiable task specifications.** By exposing an explicit formal specification layer (Reward Machines) between the human and the agent, GCR allows auditors to read, simulate, and formally verify the reward logic before deployment. This contrasts with opaque end-to-end reward models and can help regulators trace undesirable behaviour back to a concrete symbolic condition rather than a latent neural representation, supporting safer and more accountable RL pipelines.

**Broader access to capable agents.** Because symbols are grounded once and then recomposed to create an unbounded task space, domain experts without ML backgrounds can author complex tasks simply by writing RMs, potentially democratizing advanced robotics and simulation tools in education, manufacturing, and assistive settings. The same mechanism lets small-lab researchers prototype complex multi-stage tasks without the costs associated with collecting new labelled rewards.

**Reward hacking and specification gaps.** If the learned interpretation of propositions in the environment is erroneous, the resultant behaviour may no longer match human intent. In the paper, we caution that such mis-grounding can lead to harmful behaviours despite the use of a precise formal specification.

**Labour displacement.** Easier programming of general-purpose robotic skills may substitute for manual labour in logistics or assembly lines, contributing to job displacement without adequate social safety nets.

