# OpenReview forum: "Ground-Compose-Reinforce: Grounding Language in Agentic Behaviours using Limited Data"
_NeurIPS.cc/2025/Conference — NeurIPS 2025 poster_

### Official Review · Reviewer_hsjd · 2025-07-01

**Clarity:** 4
**Significance:** 3
**Originality:** 3
**Rating:** 5
**Confidence:** 3

**Summary:**

The paper introduces a data-efficient neurosymbolic framework for RL. The method is the following. First, the agent grounds a fixed vocabulary of symbols in the environment by training a labelling function on a small set of labelled trajectories; then it composes these grounded symbols using the semantics of a formal language (Reward Machines) to specify arbitrary tasks; finally it reinforces itself by generating its own reward signals and learning through standard RL. To address sparse rewards, the authors propose a compositional reward-shaping strategy. Experiments in gridworld and robotics domains show that the method generalizes out-of-distribution for new tasks, obtained by the composition of the simpler ones, encountered in the training set. The compositional reward-shaping allows for successful training.

**Questions:**

See weaknesses (1).

**Ethical Concerns:**

["NO or VERY MINOR ethics concerns only"]

**Final Justification:**

My concerns are addressed, and I rely on the authors to provide more details in the final version of the paper.

**Limitations:**

See weaknesses (2).

**Quality:**

3

**Strengths And Weaknesses:**

Strengths:
 - A clear and good introduction to the subject of Reward Machines, along with a well-defined and formalized approach description.
 - Interesting generalization results.

Weaknesses:
 - The paper would greatly benefit from additional evaluations, for example, the comparison to something like EMMA (https://proceedings.mlr.press/v139/hanjie21a.html), which is conditioned on natural language, deeper investigation into the compositional generalization ability (to what extent does it hold?), and possible cross-domain generalization, i.e., same concepts but in the different domain.
 - The limitation of the number of PVFs that should be fitted is not discussed. It is unclear how this approach scales to the open-ended real-world tasks with a large possible number of concepts.

---

> ### Author Rebuttal · Authors · 2025-07-31
>
> Thanks for your positive and thoughtful review. We appreciate that you recognized the paper’s presentation and generalization results as strengths. We believe your summary also accurately reflects the contributions of the paper.
>
> ## [“The paper would greatly benefit from additional evaluations, for example, the comparison to something like EMMA”]
>
> While works like EMMA are also concerned with teaching agents to understand language in a grounded manner, **they operate under a different set of assumptions and are not directly applicable to our problem setting**.
>
> In EMMA, the agent learns language by optimizing an external reward signal—this represents a popular strategy for building language-conditioned agents with RL [1-4]. Unfortunately, manually building a reward function that faithfully captures language under all scenarios can be challenging, and this is an important issue that our framework addresses.
>
>
> **Our framework represents an end-to-end pipeline for building language-conditioned agents** from a limited dataset $\mathcal{D}$ of labelled agent trajectories. **It does not rely on oracle access to any external, environment-specific functions** (e.g. a reward or labelling function). Instead, we ground language directly from the finite number of examples in $\mathcal{D}$.
>
> Recall that we only require access to the following to elicit behaviours:
> - a dataset $\mathcal{D}$ of agent trajectories labelled with propositional evaluations
> - a task, abstractly specified as an RM
> - an environment (without the reward function) with which the agent interacts
>
> This pipeline is inspired by agentic foundation models (e.g. vision-language-action/VLA models) that emphasize the use of existing agent-interaction data (e.g. $\mathcal{D}$) over manual design (e.g. of reward functions). However, the key advantage of our approach over VLAs is that we explicitly leverage language compositionality to demonstrate a novel form of generalization.
>
> **To provide further evidence of our claims, we have run a new baseline that more closely mimics how VLAs are trained.** Specifically, we labelled each trajectory in $\mathcal{D}$ with a corresponding description in Linear Temporal Logic (LTL) based on the propositional evaluations for that trajectory, and then trained an LTL-conditioned policy via behaviour cloning on that data. For each task in Table 1, we prompted this policy with an LTL description that closely matches the intended RM task. This baseline also failed to solve any tasks, further supporting our claim that non-compositional approaches do not exhibit this kind of generalization.
>
> **We will clarify these differences with respect to related works and add these new results to the manuscript.**
>
> [1]: Chevalier-Boisvert, et al. "BabyAI: A platform to study the sample efficiency of grounded language learning." ICLR (2019).
>
> [2]: Hermann, et al. "Grounded language learning in a simulated 3d world." arXiv:1706.06551 (2017).
>
> [3]: Hill, et al. "Grounded language learning fast and slow." ICLR (2021).
>
> [4]: Chaplot, et al. "Gated-attention architectures for task-oriented language grounding." AAAI (2018).
>
>
> ## [“deeper investigation into the compositional generalization ability (to what extent does it hold?)”]
> This is an excellent question. The advantage of using a formal language is that the compositional rules are explicit—in our case, we can systematically interpret any RM in terms of its propositional symbols. The key observation that enables compositional generalization is that we can reduce the problem of grounding the formal language of RMs (i.e. capturing how *every* possible RM should be interpreted in the environment) to grounding the finite set of propositional symbols of the language.
>
> Stated more concretely, **so long as the learned labelling function $\hat{\mathcal{L}}(s)$ represents a faithful grounding of the propositions $\mathcal{AP}$, then the rewards generated by our core framework (Algorithm 1) will faithfully reflect *any* RM task over $\mathcal{AP}$, by design**. While one can never ensure that a grounding of propositions learned from data is perfectly faithful, our experiments show that a learned labelling function $\hat{\mathcal{L}}(s)$ can indeed faithfully elicit RM behaviours in practice (when evaluated with respect to a ground-truth labelling function). This observation also informs what an ideal dataset $\mathcal{D}$ might look like in practice—it should contain a sufficient diversity of *states* to learn a robust classifier of propositional values. Crucially, it does not necessarily require a sufficient diversity of *trajectories*. This explains why our framework can faithfully elicit a wide diversity of temporally extended behaviours (expressed as RMs), even when $\mathcal{D}$ contains no successful demonstrations of such behaviours.
>
> **We will include this discussion in the paper.**
>
> ## [“The limitation of the number of PVFs that should be fitted is not discussed. It is unclear how this approach scales to the open-ended real-world tasks with a large possible number of concepts.”]
>
> The number of PVFs is linear in the number of propositional concepts, $|\mathcal{AP}|$. We agree that this may scale poorly to more open-ended environments, and in such cases, a promising strategy is to further decompose propositions, e.g. via object and relational representations. Such representations are also compositional and have been shown to be conducive to generalization (e.g. as shown in [5]).
>
> [5]: Hill, et al. "Environmental drivers of systematicity and generalization in a situated agent." ICLR (2020).

---

> > ### Comment · Reviewer_hsjd · 2025-08-05
> >
> > I thank the authors for their clarifications. My concerns are addressed and I raise the score to 5.

---

> > > ### Author Response · Authors · 2025-08-06
> > >
> > > Reviewer hsjd, thanks for your response and for raising your score.

---

### Official Review · Reviewer_J8VM · 2025-07-02

**Clarity:** 3
**Significance:** 2
**Originality:** 2
**Rating:** 4
**Confidence:** 4

**Summary:**

The proposed method inputs a set of propositional symbols and trains a labelling function to ground propositions to state using an input dataset. Given a new task expressed in formal language as a RM, the trained labeling function is then used to generate rewards for reward shaping in RL. The idea is to use provided symbols and RMs for tasks, and learn general labelling and task-specific reward functions to solve those tasks.

Contributions claims:
- Symbol grounding can be learned with predefined set of symbols and a dataset of trajectories and ground truth labels.
- The proposed reward shaping can be done with offline RL and any provided RMs (without rewards) and improves sample-efficiency and generalization in RL.

Assumptions made by the approach:
- A set of symbols or propositions are provided.
- Ground truth labelling function exists (although unobservable) so it can be used along with a dataset of trajectories to train a labelling function to ground predefined symbols to states.
- The new task is provided as a RM (without rewards but using the same predefined symbols and capturing the temporal structure of the new task).

**Questions:**

Please refer to the weakness section for questions.

**Ethical Concerns:**

["NO or VERY MINOR ethics concerns only"]

**Final Justification:**

I appreciate the authors’ response and have read the other reviews. In my view, assuming access to a reward machine and using it to generate rewards—rather than assuming access to a reward function—feels natural. However, this approach doesn’t seem sufficiently novel to me, especially since assuming the availability of a reward machine for a new task is a strong assumption. The main novelty appears to lie in learning the labeling function from data. I recommend that the authors clearly emphasize the key differences from prior reward machine approaches in the paper (how it accounts for concrete states when estimating the value of RM states). Based on these considerations, I am slightly increasing my score.

**Limitations:**

Yes

**Paper Formatting Concerns:**

No formatting concerns

**Quality:**

2

**Strengths And Weaknesses:**

Weaknesses and questions:
- The trained labelling function learns the symbol mapping for ground states. How is it also used to simulate rewards? It seems the text in the core algorithm is abstracting out necessary details - it mentions L (that returns symbols) but does not explain \delta_u and \delta_r functions used to simulate the RM states and rewards. Are \delta functions what represents given RM and so are. They assumed to be given? They need to defined and explained. Currently, these notations are not referred at all in the text.
- Currently, the novelty comes from learning a labeling function (which assumes ground truth labels for training) and learning rewards for a RM using offline RL (which assumes no self-transitions which are common in RMs). Other ideas of RMs already exist in the literature. Many reward shaping ideas also already exist (even with RMs). How is the proposed approach h different from them?
- What do you mean by unsupervised RL?
- What exact behaviors in the tasks are not present in the dataset (Table 1)?
- The empirical evaluation seems more of an ablation study. Have you tried comparing with any state-of-the-art RL approach, such as PPO [1], CAT+RL [2], Dreamer [3]?
- The related work seems cursory and needs more specific comparisons with most related works. Discuss in more detail about the methods that learn symbol grounding and explore reward shaping.

Overall assessment:

I am of the opinion that the paper is making too many assumptions across different aspects of the approach, and some of them (such as learning a RM instead of assuming it is given or removing self-transition assumption of RMs for generating rewards, etc) should be relaxed for a sufficient contribution. The novelty currently is minimal and insufficient for acceptance - training a labelling function and using provided RMs (with symbols, not rewards) to generate rewards does not seem enough. It is not clear, how the reward shaping ideas are significantly more novel compared to previous reward shaping ideas explored in the RM or LTL or other RL literature.



References:

[1] Schulman, J., Wolski, F., Dhariwal, P., Radford, A. and Klimov, O., 2017. Proximal policy optimization algorithms. arXiv preprint arXiv:1707.06347.

[2] Dadvar, M., Nayyar, R.K. and Srivastava, S., 2023, July. Conditional abstraction trees for sample-efficient reinforcement learning. In Uncertainty in Artificial Intelligence (pp. 485-495). PMLR.

[3] Okada, M. and Taniguchi, T., 2022, October. DreamingV2: Reinforcement learning with discrete world models without reconstruction. In 2022 IEEE/RSJ International Conference on Intelligent Robots and Systems (IROS) (pp. 985-991). IEEE.

---

> ### Author Rebuttal · Authors · 2025-07-31
>
> Thanks for your detailed review and questions. Below we address your questions and comments. We believe our responses address your concerns but if any issues remain, please let us know during the discussion period—otherwise, we kindly ask you to consider updating our score.
> ## [“I am of the opinion that the paper is making too many assumptions…such as learning a RM instead of assuming it is given”]
> **We assume the RM is given, not learned**. Note that we do not assume access to an external reward signal to define the agent’s objective. Instead, our goal is to task the agent directly through an RM.
>
> ## [“Are \delta functions what represents given RM and so are. They assumed to be given? They need to defined and explained. Currently, these notations are not referred at all in the text.”]
> The formal definition of an RM is in terms of $\delta_r, \delta_u$ (Definition 1 in the paper), and they govern the rewards and transitions of the RM on each step (as described on lines 89-95). Yes, they are assumed to be given as part of the RM.
>
> However, it is typically more convenient to specify an RM’s transitions via edges labelled with a logical condition and a corresponding reward (as in Figure 1). This is ultimately equivalent to specifying via $\delta_r, \delta_u$. Lines 107-109 explain how $\delta_r, \delta_u$ are defined when an RM is specified via labelled edges.
>
> ## [“removing self-transition assumption of RMs for generating rewards”]
> **Our approach does in fact address self-loop transitions in RMs**, and our experiments explicitly test this scenario in the Lift-Red-Box task.
>
> In the main text, we present Approximation 1 under the assumption of no self-loop transitions to improve the clarity of the exposition—handling self-loops involves several extra steps. This is stated on line 236, and then the reader is pointed to the extension in the Appendix that “handles any RM” on line 237.
>
> ## [“The trained labelling function learns the symbol mapping for ground states. How is it also used to simulate rewards?”]
> An RM is an abstract specification of a reward function, similar to a natural language statement that  "a reward of 1 is given for picking up a red box, then picking up a blue box". However, an RM makes the temporal structure explicit. To simulate rewards for an RM expressing this statement, we need a labelling function that, for any environment, determines whether: (1) the red box is currently picked up; (2) the blue box is currently picked up. Then, for any trajectory in the environment, once it has been observed that (1) has occurred at some time $t$, and that (2) has occurred at some time $t’ > t$, then a reward of 1 is issued at time $t’$. This last step is done by systematically updating the RM state according to $\delta_u$ and issuing rewards according to $\delta_r$.
>
> ## [“What do you mean by unsupervised RL?”]
>
> RL agents typically learn meaningful behaviours by optimizing an external reward function (i.e. the reward signal "supervises" the agent). This is also a prominent approach for building language-conditioned agents via RL—often, the agent optimizes a reward function that incentivizes the completion of any given instruction (e.g. [1-4]). These works show that agents can successfully learn to follow language instructions with a reward function that faithfully captures language, but they rarely address where this reward function comes from.
>
> Unfortunately, designing such reward functions can be notoriously hard [5]. For example, MetaWorld’s manually designed reward function for picking and placing an object: (i) involves complex logic spanning over 100 lines of Python code; (ii) requires access to internal simulator variables; (iii) targets only a single task.
>
> **The distinguishing feature of our framework is that it does not assume access to an external reward function** (or any other external, environment-specific functions e.g. a labelling function). Rather, it reflects an end-to-end pipeline for building language-conditioned agents without environment-specific design. Recall that we only assume access to the following:
> - a dataset $\mathcal{D}$ of agent trajectories labelled with propositional evaluations
> - a task, abstractly specified as an RM (similar to those shown in Figure 1)
> - an environment (without the reward function) with which the agent interacts
>
> **When we say that the agent learns via unsupervised RL, we mean that it trains via interaction with the environment, but does not have access to any external learning signal**. Instead, our problem setting requires the agent to derive its own reward signal for any given RM task based on its grounded understanding of the formal language.
>
> **Why is this setting hard?** The agent is essentially evaluating its own performance and optimizing it with RL—this runs the risk of reward hacking if the agent’s understanding of the formal language does not faithfully generalize across a wide range of scenarios.
>
> [1]: Chevalier-Boisvert, et al. "BabyAI: A platform to study the sample efficiency of grounded language learning." ICLR (2019).
>
> [2]: Hermann, et al. "Grounded language learning in a simulated 3d world." arXiv:1706.06551 (2017).
>
> [3]: Hill, et al. "Grounded language learning fast and slow." ICLR (2021).
>
> [4]: Chaplot, et al. "Gated-attention architectures for task-oriented language grounding." AAAI (2018).
>
> [5]: Amodei & Clark. Faulty Reward Functions in the Wild. https://blog.openai.com/faulty-reward-functions (2016).
>
> ## [“Other ideas of RMs already exist in the literature. Many reward shaping ideas also already exist (even with RMs). How is the proposed approach h different from them?”]
>
> Our approach differs from prior RM works because it is end-to-end—we do not require an external labelling function that (in conjunction with the RM) provides the agent’s objective. This allows us to provide a novel perspective on how to build language-based agents with minimal manual design, while only relying on a limited set of labelled agent-interaction data.
> Based on this new problem setting, we demonstrate a novel form of generalization that we argue is not exhibited by non-compositional approaches.
>
> Below, we also describe the two most relevant lines of work on RM reward shaping and how our approach differs:
> - One idea is to estimate the value of being in any RM state $u$ (without considering the environment state $s$) to generate a shaping signal (e.g. see [27-29] in the paper). Our approach improves on this by also considering $s$ when estimating the value—our contribution is how such a value function can be derived based on $\mathcal{D}$ for any RM task. We directly compare against this baseline in the experiments.
> - Another idea is to use some external signal that measures how close each proposition is to satisfaction (e.g. Manhattan distance in a gridworld) and to compose them to estimate the distance to the overall goal (e.g. [30-35] in the paper). However, these methods only apply to temporal goals with a binary success criterion. Our work’s novel contributions include: (1) showing how the initial signals can be learned directly from $\mathcal{D}$ via offline RL and (2) extending this to arbitrary RMs, which can express rich preference structures beyond binary goals.
>
> ## [“What exact behaviors in the tasks are not present in the dataset (Table 1)?”]
> - In our grid experiments, $\mathcal{D}$ consists solely of random-action trajectories, but our agent achieves near-optimal performance on a diversity of complex, temporally extended tasks.
> - In our MetaWorld-based experiments, we controlled the dataset $\mathcal{D}$ such that each trajectory interacted with at most one box, but the Pickup-Each-Box task requires the agent to pick up the red box, then the blue box, then the green box, all within the same trajectory.
> - The Lift-Red-Box task requires the agent to pick up and hold the red box as long as possible. The longest that the red box is held for in $\mathcal{D}$ is 736 steps, while our agent learns a policy that holds it for 1538 steps on average (this is close to the upper limit, since episodes last 2000 steps).
>
> Our experiments show that several non-compositional approaches fail to solve these tasks under the limited dataset $\mathcal{D}$, even when given advance knowledge of these tasks when training on $\mathcal{D}$.
>
> ## [“The empirical evaluation seems more of an ablation study.”]
>
> We are proposing a novel problem setting and to our knowledge, few existing methods are directly applicable. Thus, our experiments aim to uncover what elements enable robust language learning and generalization. Most importantly, we show the importance of exploiting the compositional syntax and semantics of the formal language.
>
> **Nevertheless, we have run a new baseline inspired by how agentic foundation models are typically trained**. Specifically, we labelled each trajectory in $\mathcal{D}$ with a corresponding description in Linear Temporal Logic (LTL) based on the propositional evaluations for that trajectory, and then trained an LTL-conditioned policy via behaviour cloning on that data. For each task in Table 1, we prompted this policy with an LTL description that closely matches the intended RM task. This baseline also failed to solve any tasks.
>
> ## [“Have you tried comparing with any state-of-the-art RL approach, such as PPO [1], CAT+RL [2], Dreamer [3]?”]
>
> **These approaches solve an orthogonal problem to our problem setting.** They are concerned with learning a policy to optimize a given reward function. Our setting does not assume access to an external reward function. Rather, learning (from $\mathcal{D}$) to generate meaningful rewards that align with any given RM task is key to the problem we are trying to solve.
>
> Note that these rewards generated by our framework can be used in tandem with any standard, model-free RL algorithm to elicit behaviours. In fact, all **RL-based methods in our experiments are optimized with PPO** (see lines 320-321).

---

> > ### Author Response · Authors · 2025-08-06
> >
> > Reviewer J8VM, we believe we addressed your main questions in our response and are looking forward to hearing back from you regarding our rebuttal. If there is anything outstanding, please let us know. We’d like to take advantage of these extra few days to respond to any remaining items.

---

> > > ### Comment · Reviewer_J8VM · 2025-08-08
> > >
> > > I appreciate the authors’ response and have read the other reviews. In my view, assuming access to a reward machine and using it to generate rewards—rather than assuming access to a reward function—feels natural. However, this approach doesn’t seem sufficiently novel to me, especially since assuming the availability of a reward machine for a new task is a strong assumption. The main novelty appears to lie in learning the labeling function from data. I recommend that the authors clearly emphasize the key differences from prior reward machine approaches in the paper (how it accounts for concrete states when estimating the value of RM states). Based on these considerations, I am slightly increasing my score.

---

> > > > ### Author Response · Authors · 2025-08-09
> > > >
> > > > Thank you for your response and for increasing your score. If accepted, we will ensure that the novel elements of our method, and of the work overall, are clearly emphasized in the camera ready.
> > > >
> > > > As a final note, we’d like to briefly touch on how RM tasks can be obtained.
> > > >
> > > > ## [“assuming the availability of a reward machine for a new task is a strong assumption”]
> > > >
> > > > We agree that specifying tasks as RMs can be more involved than natural language specification—this is a tradeoff of using a more precise specification language. However, *autoformalizing* natural language specifications into formal specifications like Reward Machines is also possible.
> > > >
> > > > **We have run an additional, small experiment showing that RMs can be reliably generated directly from natural language specifications by OpenAI’s o3 model, zero-shot.** We considered each GeoGrid task in Table 1, and prompted o3 based on a template with the following elements:
> > > > - a description of the autoformalization task (“Your job is to construct a Reward Machine
> > > > representing this reward function…”)
> > > > - the output format (“Reward Machine states should be numbered 0 , 1 , 2 , 3 , ...”)
> > > > - a description of the environment (“The environment is a gridworld…”)
> > > > - the names and descriptions of propositions (“Propositions: - red: The agent’s current cell has a red object…”).
> > > > - a description of the reward function for the desired task: (“Give a reward of 1 and terminate the episode when…”)
> > > >
> > > > We ran the model five times for consistency, and found that **o3 generated a correct RM in all trials, for all four tasks**. The generated RMs were also identical in structure to the RMs we manually constructed in each case (with the only differences being in the numbering of RM states and in the representation of equivalent logical formulas).
> > > >
> > > > **This shows that formalized task representations like RMs can often be obtained from natural language alone.** Importantly, we did not need extensive domain-specific data (e.g. trajectories from the GeoGrid agent) to accomplish this.

---

### Official Review · Reviewer_jbaf · 2025-07-03

**Clarity:** 3
**Significance:** 2
**Originality:** 3
**Rating:** 3
**Confidence:** 3

**Summary:**

This paper investigates whether diverse data is necessary for developing broad capabilities in large-scale foundation models through a data-efficient approach termed *Ground-Compose-Reinforce*. The authors propose a method that first grounds a predefined set of symbols in the environment using a labeled trajectory dataset. This grounding is performed by an agent equipped with the compositional semantics of a formal language. Once grounded, the agent can be tasked through this formal language and can learn to solve a variety of tasks by generating its own learning signals. The approach is evaluated in an image-based gridworld and a MuJoCo robotics domain, demonstrating data efficiency in training from trajectories while supporting diverse, out-of-distribution capabilities through explicit leverage of compositionality. The paper highlights the limitations of current data-driven agents in complex, safety-critical domains like robotics, where data generation is costly and out-of-distribution generalization is challenging.

**Questions:**

1. Is the dataset D in DrawerWorld manually constructed, and does it also contain failed trajectories? How are the failed trajectories constructed?
2. Can a performance comparison be made with fully data-driven large-scale foundation models?

**Ethical Concerns:**

["NO or VERY MINOR ethics concerns only"]

**Final Justification:**

Concerns remained about the paper's "data-driven" claim and comparison with data-driven models:

- A large amount of data is the core of fully data-driven approaches. The authors cannot conduct data-driven training on a limited dataset and then claim that the proposed method is superior to the data-driven paradigm.
- The fully data-driven paradigm can accommodate diverse data types, not all of which can be expressed in the form of symbols. This is also a disadvantage of this paper compared to the data-driven paradigm.

**Limitations:**

Yes.

**Paper Formatting Concerns:**

No.

**Quality:**

2

**Strengths And Weaknesses:**

### Strength

1. The authors' goal of building a data-efficient learning paradigm beyond the current mainstream data-driven large-scale foundation models is meaningful.
2. The running examples provided in the paper, such as Figure 1, are easy to understand.

### Weakness

1. The authors' key claim is that current data-driven large-scale foundation models cannot handle complex multi-stage tasks, and the algorithm designed in this paper is based on this. Although the authors have stated the reasons for this claim in the introduction, they have not compared it with fully data-driven large-scale foundation models in the experiments, which makes the motivation of the paper weak.
2. The authors state that, "However, complex multi-stage tasks like building a house remain challenging even for state-of-the-art models." In fact, some works, such as [1], have already been able to complete similar multi-stage tasks like building a house in Minecraft, and the constructed house has good functional characteristics.

[1] Guo, et al., “Luban: Building Open-Ended Creative Agents via Autonomous Embodied Verification”

---

> ### Author Rebuttal · Authors · 2025-07-30
>
> Thank you for your review. We're pleased that you found the broader goal of our work—developing agents through a more data-efficient pipeline—to be meaningful and a key strength of the paper.
>
> Below, we address each of your questions and concerns in detail. We hope our responses adequately address your points, and convey the value of our contributions. We welcome further discussion and are happy to engage during the discussion period.
>
> If we have adequately addressed your concerns, please consider raising your score. If we have not, please let us know of any remaining concerns.
>
> ## [“The authors state that, "However, complex multi-stage tasks like building a house remain challenging even for state-of-the-art models." In fact, some works, such as [1], have already been able to complete similar multi-stage tasks like building a house in Minecraft”]
>
> Thanks for raising this important point. The scope of our claim in the introduction was too broad and we will revise it to be more accurate.
>
> **However, the value of our work does not ultimately hinge on this specific claim.** Our goal of building language-driven agents through a more data-efficient pipeline is important for domains where agent-interaction data is limited or expensive. This applies to many important domains, such as those that require real-world interactions (e.g. robots) [1], those that are safety-critical, and those that involve atypical observation and action spaces [2]. Prior evidence suggests that standard approaches for training foundation models are not well-suited in these data-impoverished settings [3].
>
> As MineCraft does not exemplify such a domain (there exist massive and diverse datasets of gameplay generated by human players [4, 5]) we will remove it as a motivating example.
>
> [1]: Levine. https://sergeylevine.substack.com/p/sporks-of-agi (2025).
>
> [2]: Bommasani, et al. "On the opportunities and risks of foundation models." arXiv:2108.07258 (2021).
>
> [3]: Kaplan, et al. "Scaling laws for neural language models." arXiv:2001.08361 (2020).
>
> [4]: Guss, et al. "MineRL: A large-scale dataset of minecraft demonstrations." IJCAI (2019).
>
> [5]: Baker, et al. "Video pretraining (VPT): Learning to act by watching unlabeled online videos." NeurIPS (2022).
>
>
> ## [“Can a performance comparison be made with fully data-driven large-scale foundation models?”]
>
> We do not compare with existing large-scale foundation models for a few reasons:
>
> - **It is not relevant to the research question at hand.** Our goal is to explore whether we can build language-based agents in settings with limited agent-interaction datasets—and furthermore, whether they can generalize to behaviours beyond this data. Comparing with an existing model that has already been trained on large-scale data does not inform our answer as it violates the premise of our question while making it hard to assess generalization.
>
> - **Our domains are customized.** DrawerWorld is a modified version of MetaWorld with additional objects. This diversifies the possible behaviours in the environment to help us test multi-task capabilities and generalization, but it also means that our observation space is atypical. Similarly, GeoGrid observations are encoded as 8x8x6 arrays in a non-standard format.
>
> However, we agree that comparing our approach with the training techniques of these foundation models under equivalent training data conditions is appropriate and informative.
>
> **Thus, we have run an additional comparison with a formal-language-conditioned behaviour cloning baseline.** To train this baseline, we labelled each trajectory in our trajectory dataset ($\mathcal{D}$ in the paper) with a corresponding description in Linear Temporal Logic (LTL) based on the propositional evaluations for that trajectory, and then trained an LTL-conditioned policy via behaviour cloning on that data. For each task in Table 1, we prompted this policy with an LTL description that closely matches the intended RM task. This baseline failed to solve any tasks, roughly matching the performance of “End-to-End Behaviour Cloning”. We will add these details to our revised manuscript.
>
> Note that the original baselines in our paper, End-to-End Reward Model and End-to-End Behaviour Cloning, already support our claim that non-compositional approaches fail to generalize to significantly out-of-distribution tasks given limited training trajectories in $\mathcal{D}$. However, this new baseline provides a closer comparison to how agentic foundation models (e.g. vision-language-action models) are usually trained.
>
>
> ## [“Is the dataset D in DrawerWorld manually constructed”]
>
> The dataset $\mathcal{D}$ in DrawerWorld was constructed by one of the authors manually controlling the robot via keyboard inputs. To ensure a diverse set of scenarios, the initial configuration of the environment was randomized (in terms of locations of boxes, and whether each drawer was open or closed), and the robot was controlled to interact with drawers and boxes (e.g. moving boxes into or out of drawers). In total, 350 trajectories were collected, and importantly, none of these trajectories involved direct interactions with multiple boxes. This allowed us to evaluate out-of-distribution generalization to picking up multiple boxes within the same episode. **We will add these details, as well as those in the response to follow, to the paper or appendix.**
>
>
> ## [“does it also contain failed trajectories?”]
>
> The data includes accidental behaviours such as failing to grip a box, dropping a box when attempting to move it, and opening a drawer beyond its limit. We also included behaviours not tied to downstream tasks, such as placing boxes on top of a drawer, or throwing a box off the table.
>
> The inclusion of these behaviours in $\mathcal{D}$ improves the robustness of models trained on $\mathcal{D}$. For example, we found that only including successful demonstrations leads to increased reward hacking by the downstream RL agent—the agent might deliberately reach out-of-distribution states (with respect to $\mathcal{D}$), because the model’s outputs are poorly calibrated on such states and often result in erroneously high rewards.
>
> This presents an interesting problem in itself, but is outside the scope of this paper. For this work, we simply assume the dataset $\mathcal{D}$ covers a sufficiently diverse set of states to robustly capture the propositions we care about (e.g. whether a drawer is open, or a box is picked up). Nevertheless, as our experiments show, we are still able to demonstrate generalization to tasks significantly out-of-distribution with respect to $\mathcal{D}$ (e.g. picking up all three boxes within the same episode, or holding a box as long as possible).

---

> > ### Comment · Reviewer_jbaf · 2025-08-05
> >
> > Thank you to the authors for their response, which has addressed most of my concerns.
> >
> > The authors argue that fully data-driven approaches are difficult to generalize beyond the training data. However, research over the past two years on large foundation models (such as LLMs and VLMs) has shown that existing large foundation models do have good generalization capabilities. Although the authors added a fully data-driven baseline in the rebuttal, it is clear that with such a small amount of data, this baseline would not have any generalization ability. Therefore, demonstrating some generalization ability with limited data is indeed an advantage of the method presented in this paper. However:
> >
> > - A large amount of data is the core of fully data-driven approaches. You cannot conduct data-driven training on a limited dataset and then claim that your method is superior to the data-driven paradigm.
> > - The fully data-driven paradigm can accommodate diverse data types, not all of which can be expressed in the form of symbols. This is also a disadvantage of this paper compared to the data-driven paradigm.
> >
> > Therefore, I am willing to raise my score from 2 to 3.

---

> > > ### Author Response · Authors · 2025-08-07
> > >
> > > Thanks for engaging, for your response, and for updating your score. We’re glad to hear that most of your concerns have been addressed, and that you recognize the ability of our method to generalize from limited data as an advantage. We briefly address your outstanding concerns below.
> > >
> > > ## [“A large amount of data is the core of fully data-driven approaches. You cannot conduct data-driven training on a limited dataset and then claim that your method is superior to the data-driven paradigm.”]
> > >
> > > **Important: We do not claim to be superior to the “data-driven paradigm”, nor do we dispute that a greater availability of data is advantageous (as evidenced by scaling laws).**
> > >
> > > Our work is motivated by settings where data is inherently limited or expensive—where the “data-driven paradigm” is neither a practical nor an economical option. For example, this case might arise if a small company designs a new robot with non-standard sensors and actuators (as is often the case in robotics, per [2] in the previous response) and has a limited budget for generating and labelling bespoke trajectories from this robot.
> > >
> > > As you remark (and as our new experiments show), typical methods for training agentic foundation models do not fare well in the data-limited regime—hence why we believe our contributions to be timely and important.
> > >
> > > ## [“The fully data-driven paradigm can accommodate diverse data types, not all of which can be expressed in the form of symbols. This is also a disadvantage of this paper compared to the data-driven paradigm.”]
> > >
> > > **To be clear, our framework does accommodate “diverse data types” in the sense that the agents can have virtually any observation and action space.** However, our goal in this work is to develop agents that can also be prompted through language (specifically, through the formal language of RMs).
> > >
> > > While we agree that prompting an agent via RMs requires the desired behaviour to be expressible through a composition of symbols, **we do not consider this to be a significant disadvantage compared to other approaches**. Every language—including natural language, programming languages, and domain-specific languages (all of which are extensively used for tasking agents)—captures meaning through symbols. Therefore, data-driven agents that are prompted via language, including vision-language-action models and chatbots, also rely on the expression of behaviours through symbols.
> > >
> > > If your concern is with the expressivity of the particular symbol system in our work, **note that these symbols can represent virtually anything of a propositional nature (i.e. that has a true or false outcome)**.
> > >
> > > Finally, your concern may be that our system cannot also be prompted via non-language modalities—e.g. perhaps you wish to prompt the agent with an image of the house you want it to build. However, you could accomplish this by including the additional prompt modality in the agent’s observations and then defining propositions based on that modality (e.g., the proposition that the shape of the constructed house matches the shape of the house in the prompt image). Nonetheless, the main focus of our work is the integration of agents with language.
> > >
> > > [6] Sun, Wu, and Lim. "Program guided agent." ICLR 2020.

---

### Official Review · Reviewer_9UHD · 2025-07-04

**Clarity:** 2
**Significance:** 2
**Originality:** 2
**Rating:** 5
**Confidence:** 3

**Summary:**

This paper is a follow-up on the Reward Machines line of research. There are two key innovations to the work with respect to previous papers. First, it learns the label function from data, so it relies on what is true in a reward machine on this learned function. Second, it uses the steps within an RM as subgoals to derive dense reward signals. The method is evaluated on a gridworld problem and a Mujoco task. The results show that the learned label function with a subgoal-based learned reward function outperforms baselines that are variations of the proposed method.

**Questions:**

- Could you please explain Equation 1 to me? Why is this a reasonable approximation?
- What are the novel pieces of the system?

**Ethical Concerns:**

["NO or VERY MINOR ethics concerns only"]

**Final Justification:**

I thank the authors for answering my questions. I better understand the novelty of the contribution and the intuition behind Equation 1. I am thus raising my score.

**Limitations:**

The paper addresses the main limitations.

**Paper Formatting Concerns:**

No concerns with formatting.

**Quality:**

2

**Strengths And Weaknesses:**

Strengths:

- The direction of combining formal languages and RL is quite compelling, so I like the general direction of this work.
- One of the key issues with this type of neurosymbolic method is the amount of human-annotated data, and this work represents a small step toward streamlining the use of RM in practice.
- The idea of using the transitions of the RM to generate a denser reward function is also interesting and valuable, although I am not entirely sure how novel it is.
- The paper has one of the best RM presentations that I have seen. The writing flows smoothly, with examples sprinkled throughout - nicely done!

Weaknesses:

- The experiments presented in the paper are on the weaker side because they use only five random seeds, which is often insufficient to draw conclusions in RL experiments. The paper presents only a table with the return results, while learning curves could also be informative.
- The paper doesn't provide any rationale for Equation 1. I don't understand why $V*$ multiplies the sum $r + v_R*$. Why would the return of achieving a subgoal be multiplied by the reward observed in one time step, added to the stateless value function? This makes no sense to me. I went to the appendix to try to understand this equation, but still failed, as I don't understand the step from $\gamma$ to $V^*$.
- The paper is of moderate novelty at best.

---

> ### Author Rebuttal · Authors · 2025-07-31
>
> Thanks for your positive evaluation of our work. We appreciate your recognition of the quality of our presentation, and we’re glad that you found the examples in the paper useful.
>
> An important contribution that we’d like to reemphasize is our novel problem setting—we believe this work provides a new and compelling perspective on how to build language-based agents in domains with limited data. Below, we explain the advantages of our high-level framework over current approaches and why we think our insights are valuable to a broader community.
>
> We hope our response clarifies the value of this work, and we look forward to a productive discussion period.
>
> ## [“What are the novel pieces of the system?”]
>
> We agree that each component of our approach is based on prior techniques (though our reward shaping approach has key differences from previous works, as we later explain). However, the aggregation of these components solves an important problem in a novel way that yields advantages over prior approaches.
>
> Specifically, **our primary contribution in this work is an end-to-end framework** that takes a limited dataset $\mathcal{D}$ of labelled agent trajectories and, via (formal) language prompting, elicits a wide range of useful behaviours. **This is achieved without significant manual engineering or oracle access to any external, environment-specific functions** (e.g. a reward or labelling function). Recall that we only require access to the following:
>
> - a dataset $\mathcal{D}$ of agent trajectories labelled with propositional evaluations
> - a task, abstractly specified as an RM
> - an environment (without the reward function) with which the agent interacts
>
> **Why is this different from prior works in RL?** To build a language-conditioned agent, one must capture a complex mapping from language to states, actions, and behaviours in the environment. Language-driven RL typically assumes access to a reward function that captures this mapping by incentivizing the completion of any given language task (e.g. [1-4]). Similarly, formal-language-based RL (e.g. with RMs or LTL) assumes access to a labelling function, from which rewards are derived automatically. These approaches are shown to perform well when the reward or labelling function faithfully reflects language under all scenarios, but they rarely address where such a function comes from.
>
> Unfortunately, designing such reward/labelling functions can be notoriously hard [5]. For example, MetaWorld’s manually designed reward function for picking and placing an object: (i) involves complex logic spanning over 100 lines of Python code; (ii) requires access to internal simulator variables; (iii) targets only a single task.
>
> Our end-to-end approach captures complex and meaningful MetaWorld behaviours from only a modest dataset $\mathcal{D}$ of 350 labelled trajectories. We systematically construct dense reward functions that: (i) require no environment-specific design; (ii) are based on the agent’s raw observations; (iii) can target any RM task over the set of propositional symbols.
>
> **Why is this different from agentic foundation models (e.g. vision-language-action models/VLAs)?** Our end-to-end framework builds language-based agents in a manner that is inspired by VLAs—with emphasis on leveraging existing trajectory data (i.e. $\mathcal{D}$), and with minimal manual design. However, the key advantage of our approach over VLAs is that we explicitly leverage language compositionality to demonstrate a novel form of generalization.
>
> We designed our experiments with controlled datasets $\mathcal{D}$ to evaluate generalization to behaviours that are significantly out-of-distribution with respect to $\mathcal{D}$. In GeoGrid, $\mathcal{D}$ is comprised solely of random-action trajectories, but our agent achieved near-optimal performance on a diversity of complex, temporally extended tasks. In our MetaWorld-based domain, each trajectory in $\mathcal{D}$ interacts with at most one box, but our agent learned to reliably pick up all three boxes within the same trajectory.
>
> To demonstrate why this kind of generalization is hard, we compared against non-compositional approaches that learn from $\mathcal{D}$. These methods failed to reliably solve these out-of-distribution tasks, despite having advance knowledge of the tasks when training on $\mathcal{D}$.
>
> **To provide further evidence, we have run a new baseline that more closely mimics how VLAs are trained.** Specifically, we labelled each trajectory in $\mathcal{D}$ with a corresponding description in Linear Temporal Logic (LTL) based on the propositional evaluations for that trajectory, and then trained an LTL-conditioned policy via behaviour cloning on that data. For each task in Table 1, we prompted this policy with an LTL description that closely matches the intended RM task. This baseline also failed to solve any tasks, further supporting our claim that non-compositional approaches do not exhibit this kind of generalization.
>
> **We will clarify these points of novelty and add our new results to the manuscript.**
>
> [1]: Chevalier-Boisvert, et al. "BabyAI: A platform to study the sample efficiency of grounded language learning." ICLR (2019).
>
> [2]: Hermann, et al. "Grounded language learning in a simulated 3d world." arXiv:1706.06551 (2017).
>
> [3]: Hill, et al. "Grounded language learning fast and slow." ICLR (2021).
>
> [4]: Chaplot, et al. "Gated-attention architectures for task-oriented language grounding." AAAI (2018).
>
> [5]: Amodei & Clark. Faulty Reward Functions in the Wild. https://blog.openai.com/faulty-reward-functions (2016).
>
>
> ## [“using the transitions of the RM to generate a denser reward function is also interesting and valuable, although I am not entirely sure how novel it is.”]
>
> Prior works exploit continuous “distance metrics” towards the satisfaction of propositions (references [30-35] in the paper). However, our approach is different in two ways:
> - Prior works typically manually design these distance metrics. Our entire framework operates without relying on such manual design, and we instead show how these metrics can be learned directly via offline RL techniques from $\mathcal{D}$.
> - Prior works focus on satisfaction of a formal specification $\varphi$ (i.e. with a binary success criterion) and estimate a notion of distance to satisfying the overall $\varphi$. However, it’s unclear how to apply their techniques to general RMs, which are not strictly limited to goals of a binary nature. Inspired by [27] in the main paper, our approach is based on estimating the optimal value function for any RM.
>
> **We will make these differences more clear in the related works section.**
>
> ## [“The paper presents only a table with the return results, while learning curves could also be informative.”]
>
> Due to space limitations, the learning curves are currently found at the end of the Appendix (Figures 5, 6) but are referenced in the main text on line 326 and 333-336.
>
> Because of our unique setting, where part of the challenge is to learn to generate useful rewards based on the dataset $\mathcal{D}$, the learning curves show performance w.r.t. both these learned rewards as well as a ground-truth notion of rewards.
>
> ## [“The experiments presented in the paper are on the weaker side because they use only five random seeds”]
>
> We agree that more seeds would strengthen the results. The number of seeds is informed by a few factors:
> - Computational cost: Our RL domains are challenging and require up to 20M frames of training. The computational cost is exacerbated by inference costs from the labelling function and PVF networks as well as the internal logic of the RM. These calls must be made on every step of the environment.
> - Stability of results: The error bars in the learning curves and the table already demonstrate consistent performance across seeds.
>
> We believe five seeds balances the robustness of our claims with computational cost, in line with RL community standards.
>
> ## [“Could you please explain Equation 1 to me?”]
>
> Note that there is a minor mistake in Equation 1: the term inside the maximization should read $V^{\ast}_{\Diamond \varphi}(s) \cdot (r + \gamma v^{\ast}\_\mathcal{R}(u'))$ (however, based on your comment, this doesn’t appear to be the source of confusion).
>
> Recall that we estimate the return for immediately satisfying the RM transition $\langle u, u’, \varphi, r \rangle$ as $r + \gamma v^\ast\_{\mathcal{R}}(u’)$.  $r$ is immediately received from the RM transition and $v^\ast\_{\mathcal{R}}(u’)$ estimates the future discounted return for being in RM state $u’$. Again, we believe this part is not under dispute.
>
> Suppose that the number of steps it takes to satisfy $\varphi$ (which causes the RM transition) from the agent’s current state $s$ is $K$ (where $K$ is a random variable that could be $\infty$ if the RM transition never occurs). The delayed return for satisfying $\varphi$ $k \sim K$ steps in the future is $\gamma^k ( r + \gamma v^\ast\_{\mathcal{R}}(u’) )$, or in expectation, $\mathbb{E}\_{k \sim K}[\gamma^k ( r + \gamma v^\ast\_{\mathcal{R}}(u’) )]$. Since the second factor is constant w.r.t. $k$, we can rewrite this as $(\mathbb{E}\_{k \sim K}[\gamma^k]) \cdot ( r + \gamma v^\ast\_{\mathcal{R}}(u’) )$.
>
> To maximize this expected discounted return, the agent ought to maximize $\mathbb{E}\_{k \sim K}[\gamma^k]$ (which balances the probability of eventually satisfying $\varphi$ with satisfying it quickly). This is precisely the expected discounted return from state $s$ for the task $\Diamond \varphi$ (which yields a reward of 1 on the timestep $\varphi$ is satisfied), hence the maximum value of $E_{k \sim K}[\gamma^k]$ is $V^*_{\Diamond \varphi}(s)$, which results in our desired expression.
>
> We hope this clarifies Equation 1 and we will add this explanation to the Appendix.

---

### Note · Authors · 2025-08-12

Dear AC and Reviewers,

We sincerely thank you for the constructive feedback and productive discussion.

**For the AC:**

*Our main contribution is an end-to-end framework for building multi-purpose agents that can be prompted via formal language.* This integrates ideas from two distinct areas—data-driven agentic systems (e.g. vision-language-action models) and formal-language-based RL—but yields meaningful advantages over both.
- We do not require domain-specific, manually engineered oracles (e.g. reward functions or labelling functions).
- By exploiting language compositionality, we can faithfully elicit behaviours given only a modest dataset of labelled trajectories, while compositionally generalizing beyond that dataset.

**The reviewers highlighted the following strengths of our work:**
- Reviewers 9UHD and jbaf found our broader goal of building agents in a data-efficient manner to be well justified.
- Reviewer hsjd highlighted our “interesting” generalization results as a strength.
- Most reviewers praised the presentation. Reviewer 9UHD called it “one of the best RM presentations that I have seen”, Reviewer hsjd described it as “a clear and good introduction to the subject of Reward Machines”, and Reviewer jbaf found the running examples “easy to understand”.

**The following issues were addressed in rebuttal (and will be reflected in the camera ready, if accepted):**

- **Novelty:** We clarified our primary contribution (as noted above) and its novelty. We believe our framework provides a new perspective on how to build language-driven agents with less labelled trajectory data.
- **VLA-inspired baseline:** We ran a new baseline based on the training procedure of agentic foundation models (e.g. VLAs) in response to Reviewer jbaf. The results support our initial claim: non-compositional approaches fail to exhibit robust grounded language understanding or generalization in a data-limited regime.
- **Prompting via natural language:** We ran a small experiment (in response to reviewer J8VM) to show that the agent in our framework can be tasked directly via natural language using an autoformalizer. We found that our RMs can be reliably generated zero-shot by OpenAI’s o3 from natural language specifications alone. More broadly, this shows how formal representations can be an effective, under-the-hood tool even for agents interfaced via natural language.

We believe these changes will help clarify the novelty and broader relevance of our work.

---

### Decision · Program_Chairs · 2025-09-17

**Decision:**

Accept (poster)

**Comment:**

The paper introduces ground-compose-reinforce, a neurosymbolic framework for tasking rl agents via formal language. Symbols are grounded from labeled trajectories, composed using reward machines to specify complex tasks, and reinforcement is achieved by exploiting compositional structure to self generate learning signal. A compositional reward shaping strategy predicts value functions for arbitrary tasks, enabling efficient learning in settings of sparse rewards. Exps in gridworld and mujoco show data-efficient training + generalization to diverse, OOD tasks beyond the original trajectories .

In the original round of reviews the committee appreciated the clear and formalized presentation of the reward machines framework and the strong exposition with intuitive examples (reviewers 9UHD, hsjd), the direction of combining formal lang. with rl for data efficiency and compositional generalization (R#jbaf, R#hsjd), and the ability to elicit diverse out-of-distribution behaviors from small trajectory datasets (R=9UHD,J8VM). The overall technical solidity and potential impcat of the neurosymbolic approach was recognized.

During rebuttal and discussion, the authors clarified the novelty of their framework, explained eq. 1 in detail, and added new baselines mimicking foundation-model training, which supported their generalization claims. R-9UHD’s concerns on equation rationale and novelty were resolved, leading to a higher score; R-jbaf acknowledged the advantage in data-limited regimes and raised their score despite comparisons to large-scale models; R-J8VM appreciated clarifications also raised their score; reviewer hsjd’s requests on evaluation and scaling were addressed, and they raised their score to 5. The AC concurs with the majority of the committee’s reviews and post-rebuttal discussions that lean towards acceptance. The authors are encouraged to incorporate constructive discussion in a revised version of the paper.